# TIR-1/SARM1 inhibits axon regeneration and promotes axon degeneration

**Victoria L Czech†, Lauren C O'Connor†, Brendan Philippon, Emily Norman, Alexandra B Byrne\***

Department of Neurobiology, UMass Chan Massachusetts Medical School, Worcester, United States

**Abstract** Growth and destruction are central components of the neuronal injury response. Injured axons that are capable of repair, including axons in the mammalian peripheral nervous system and in many invertebrate animals, often regenerate and degenerate on either side of the injury. Here we show that TIR-1/dSarm/SARM1, a key regulator of axon degeneration, also inhibits regeneration of injured motor axons. The increased regeneration in *tir-1* mutants is not a secondary consequence of its effects on degeneration, nor is it determined by the NADase activity of TIR-1. Rather, we found that TIR-1 functions cell-autonomously to regulate each of the seemingly opposite processes through distinct interactions with two MAP kinase pathways. On one side of the injury, TIR-1 inhibits axon regeneration by activating the NSY-1/ASK1 MAPK signaling cascade, while on the other side of the injury, TIR-1 simultaneously promotes axon degeneration by interacting with the DLK-1 mitogen-activated protein kinase (MAPK) signaling cascade. In parallel, we found that the ability to cell-intrinsically inhibit axon regeneration is conserved in human SARM1. Our finding that TIR-1/SARM1 regulates axon regeneration provides critical insight into how axons coordinate a multidimensional response to injury, consequently informing approaches to manipulate the response toward repair.

**\*For correspondence:**
alexandra.byrne@umassmed.edu

†These authors contributed equally to this work

**Competing interest:** The authors declare that no competing interests exist.

## Editor's evaluation

This important study reveals a new role for TIR-1/SARM1 in inhibiting both axonal regeneration and promoting axon degeneration in a novel axonal injury model. The authors provide solid evidence to support their conclusions that TIR-1 mediates these processes by acting via distinct signaling pathways. This work will be of general interest to developmental neuroscientists.

## Introduction

In many injured neurons that are capable of repair, axons are broken into fragments with different fates. The axon segment that remains attached to the cell body regenerates and severed axon segments degenerate. However, in the central nervous system and in aged animals, regeneration of the proximal axon fragment is inhibited, resulting in a permanent loss of neuronal function (*Cai et al., 2001*; *David and Aguayo, 1981*). Understanding the molecular mechanisms that regulate both axon regeneration and degeneration is critical to developing therapeutic approaches to repair the injured nervous system.

On the proximal side of the injury, functional axon regeneration requires the synchronized execution of multiple processes, including growth cone formation, axon guidance, and synapse formation. Despite being recapitulations of well-characterized developmental processes, the precise molecular mechanisms that regulate axon regeneration remain incompletely understood. Those that have been identified are subdivided into two broad categories: intrinsic and extrinsic. Intrinsic regulators

of transcription, transport, translation, and signaling are complemented and opposed by extrinsic regulators of myelin-dependent inhibition and scar formation (*He and Jin, 2016*; *Kaplan et al., 2015*; *Mahar and Cavalli, 2018*; *Schwab and Strittmatter, 2014*; *Silver et al., 2014*; *Sutherland and Geoffroy, 2020*). Manipulating either its intrinsic or extrinsic environment is sufficient to modestly increase the ability of an injured axon to regenerate (*Curcio and Bradke, 2018*; *He and Jin, 2016*; *Schwab and Strittmatter, 2014*; *Tedeschi and Bradke, 2017*; *Tran et al., 2018*).

On the distal side of the injury, the most notable endogenous regulator of axon degeneration is Sterile Alpha and toll/interleukin-1 resistance (TIR) motif containing 1 (SARM1/dSarm), which is essential for injury-induced axon degeneration of mouse and fly axons (*Gerdts et al., 2013*; *Osterloh et al., 2012*). SARM1/dSarm promotes axon degeneration cell-autonomously by depleting NAD$^+$, interacting with the BTB/BACK domain protein Axundead, and interacting with mitogen-activated protein kinase (MAPK) signaling cascades, which in turn regulate calpain activation, SCG10/Stathmin-2 proteolysis, mitochondrial dysfunction, cytoskeletal disruption, and axon fragmentation (*Coleman and Höke, 2020*; *Essuman et al., 2017*; *Gerdts et al., 2015*; *Gerdts et al., 2016*; *Neukomm et al., 2017*; *Shin et al., 2012*; *Summers et al., 2020*; *Walker et al., 2017*; *Yang et al., 2015*).

The *Caenorhabditis elegans* SARM1/dSarm homolog, TIR-1, was first characterized as a determinant of neuronal asymmetry and the innate immune response, which it regulates in coordination with the NSY-1/ASK1 MAPK signaling pathway (*Chuang and Bargmann, 2005*; *Couillault et al., 2004*; *Liberati et al., 2004*). In the absence of injury, TIR-1 has been found to regulate neurodegeneration in response to disease and age, upon multimerization, and in the absence of mitochondria (*Ding et al., 2022*; *Lezi et al., 2018*; *Loring et al., 2021*; *Vérièpe et al., 2015*); however, whether TIR-1 regulates injury-induced axon degeneration is not known.

Here we report that in addition to promoting injury-induced axon degeneration, TIR-1/SARM1 functions cell-autonomously to inhibit axon regeneration. Specifically, we find that loss of TIR-1 increases axon regeneration and that the ability to inhibit axon regeneration cell-intrinsically is conserved in human SARM1. In response to injury, TIR-1 functions with the NSY-1/ASK1 MAPK signaling cascade and the terminal transcription factor DAF-19/RFX to inhibit regeneration of the proximal axon fragment. In parallel, TIR-1 functions independently from NSY-1/ASK1 signaling and with the DLK-1/DLK MAPK pathway to promote degeneration of the distal axon fragment. Together, our results indicate that TIR-1 inhibits axon regeneration cell-autonomously, and that it does so by regulating a downstream pathway that is separable from its role in degeneration. Consequently, our results reveal a mechanism to shift the injury response toward repair.

## Results

### TIR-1/SARM1 inhibits axon regeneration

In a candidate screen for genes that function on both sides of an injured axon, we asked whether *tir-1*, the highly conserved *C. elegans* homolog of *Sarm1/dSarm* (*Mink et al., 2001*), regulates the injury response (*Figure 1A*). To do so, we severed GABA motor neurons in *tir-1* loss-of-function and wild-type animals with a pulsed laser and compared their morphology 24 hr later. We tested two putative null alleles of *tir-1*; the *qd4* allele is a 1078 base pair deletion spanning exons 8–10 and the *tm3036* allele is a smaller 269 base pair deletion in exon 8 that results in an early stop codon (*Pujol et al., 2008*; *Shivers et al., 2009*). Both the *qd4* and *tm3036* deletions disrupt the C-terminal toll/interleukin-1 receptor (TIR) domain, which is essential for TIR-1 function (*Chuang and Bargmann, 2005*; *Figure 1A*). Strikingly, we found that when GABA motor axons of *tir-1* loss-of-function and wild-type animals were severed at the midline, significantly more axons regenerated in the absence of *tir-1* (*Figure 1B–D*). These data reveal that TIR-1 inhibits motor axon regeneration.

*C. elegans* have one other TIR domain-containing protein, which is TOL-1 (*Liberati et al., 2004*). To determine whether regulation of axon regeneration is a general function of TIR domain-containing proteins, we compared regeneration in the presence and absence of *tol-1*. We found no significant difference between the number of axons that regenerated in *tol-1(nr2033)* null mutants and wild-type animals (*Figure 1—figure supplement 1*). Therefore, TIR-1 is a TIR-domain-containing protein with a specific role in axon regeneration.

Successful regeneration of *C. elegans* motor axons includes growth cone formation, axon extension, and responsiveness to cues that guide the axon to the dorsal nerve cord. To determine which

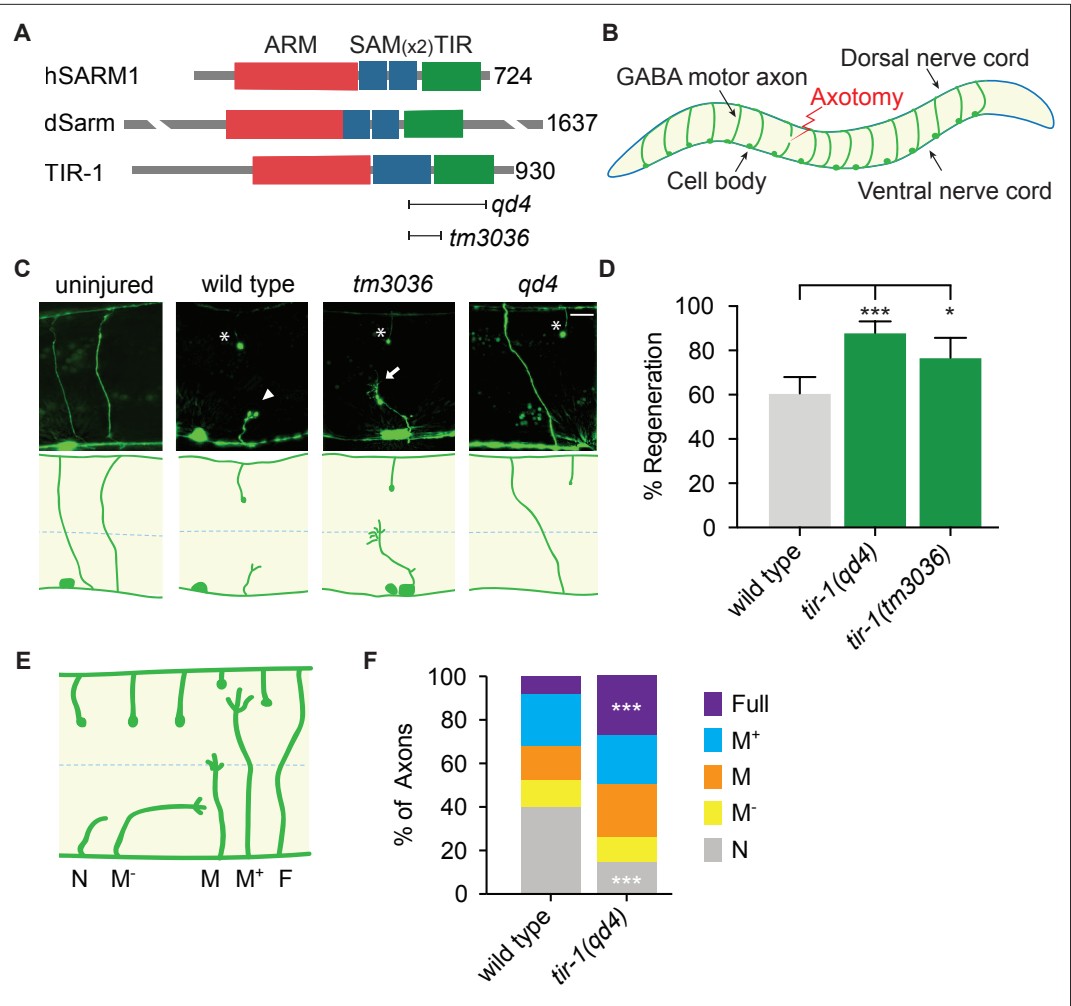

**Figure 1.** TIR-1 inhibits axon regeneration. (**A**) Human SARM1, fly dSarm, and worm TIR-1 contain an N-terminal auto-inhibitory domain, two sterile alpha motif (SAM) domains and a toll-interleukin-1 receptor (TIR) domain. Null alleles *qd4* and *tm3036* disrupt the essential TIR domain. (**B**) *C. elegans* GABA motor neurons are axotomized to study the injury response in vivo and with single-axon resolution. (**C**) Representative GABA motor neurons before and 24 hr after single laser injury at the lateral midline of late larval stage (L4) animals (dashed line). Asterisks indicate distal fragments, arrowheads indicate retraction bulbs, and arrows indicate growth cones. Scale bar: 10 μm. (**D**) More axons regenerate in *tir-1* mutants compared to wild-type animals, as quantified by the percentage of axons that form a growth cone 24 hr after laser surgery. N = 152, 99, 60. (**E**) Regeneration phenotypes were categorized according to whether regenerating axons reached landmarks M⁻, M, M⁺, F, or failed to form a growth cone, N. (**F**) More *tir-1*(-) axons initiate regeneration and reach the dorsal cord compared to wild-type axons, as indicated by a significant reduction in the number of axons that failed to form a growth cone (N) and a significant increase in the number of fully regenerated axons (Full). N = 158, 154. Significance relative to wild type is indicated by *$p \leq 0.05$, ***$p \leq 0.001$, Fisher's exact test. Error bars represent 95% confidence intervals.

The online version of this article includes the following figure supplement(s) for figure 1:

**Figure supplement 1.** TOL-1 does not regulate axon regeneration.

stage of the regenerative process is inhibited by TIR-1, we grouped individual injured axons into categories depending on whether they initiated a growth cone and how far they regenerated toward the dorsal nerve cord (**Figure 1E and F**). We found a dramatic decrease in the number of axons that did not initiate regeneration (N) and an increase in the number of axons that regenerated the full distance to the dorsal nerve cord (Full) in *tir-1* mutants compared to wild-type animals (**Figure 1F**). Of note, *C. elegans* are capable of repairing injured mechanosensory axons both by regenerating and fusing severed proximal and distal axon fragments (**Ghosh-Roy et al., 2010**; **Neumann et al., 2011**). However, we did not observe fusion of severed motor axon fragments in either wild-type or *tir-1*

mutant animals. Together, these results indicate TIR-1 inhibits the initiation of axon regeneration and the successful extension of regenerating axons to their targets.

## TIR-1 functions cell-autonomously in GABA motor neurons to regulate axon regeneration

We next asked how TIR-1 regulates axon regeneration and how it achieves specificity of function. TIR-1 functions in multiple cell types, including the nervous system, intestine, and epidermis, where it regulates neuronal development and the innate immune response (*Chuang and Bargmann, 2005*; *Couillault et al., 2004*; *Liberati et al., 2004*; *Pujol et al., 2008*). To determine where TIR-1 functions to regulate axon regeneration, we first CRISPR tagged the endogenous *tir-1* locus with mCherry and analyzed its expression pattern. With this single-copy integration of mCherry, we observed faint TIR-1 expression in the nervous system, particularly in the nerve ring, select cells in the head and tail, and in the dorsal cord and ventral cord, as well as in the intestine (*Figure 2A*). While tagging the endogenous locus is perhaps the most accurate in vivo representation of a gene's expression pattern, cells that express the gene at low levels may not be detected. Therefore, we also constructed a transgenic strain expressing multiple copies of *tir-1b::mCherry* under the endogenous *tir-1* promoter. The *tir-1b* isoform encodes the SAM and TIR domains but not the autoinhibitory N-terminus of TIR-1, and is predicted to be an activated form of TIR-1 (*Chuang and Bargmann, 2005*). With this strain, we observed a similar pattern of TIR-1 expression in the intestine and nervous system. However, the increased copy number revealed TIR-1b::mCherry expression in the cell bodies and axons of GABA motor neurons (*Figure 2B*). The finding that TIR-1 is expressed in GABA motor neurons, albeit at relatively low levels, raises the possibility that TIR-1 could function cell-intrinsically to regulate axon regeneration.

We investigated where TIR-1 functions to regulate regeneration by asking whether tissue-specific expression of *tir-1b* cDNA is sufficient to inhibit axon regeneration of injured GABA motor neurons. We expressed a GABA-specific *unc-47p::mScarlet::tir-1b* construct in *tir-1(qd4)* animals and observed punctate mScarlet::TIR-1b in the cell bodies and axons of GABA neurons, both before and after injury (*Figure 2C and D*). Axon regeneration was fully suppressed in animals that expressed *tir-1b* specifically in their GABA neurons compared to *tir-1(qd4)* animals (*Figure 2E*). Therefore, cell-intrinsic *tir-1* function inhibits axon regeneration.

The significance of the finding that *tir-1* regulates regeneration is influenced by whether its function is conserved in homologous *Sarm* genes. TIR-1 and human SARM1 proteins are highly conserved, sharing 60% homology over 85% of the SARM1 sequence (*Altschul et al., 1997*; *Mink et al., 2001*). As a result, both TIR-1 and human SARM1 contain multiple HEAT/Armadillo repeats in the autoinhibitory N-terminus followed by two sterile alpha motifs (SAM) and one toll-like interleukin repeat (TIR) domain (*Figure 1A*). We investigated whether intrinsic inhibition of axon regeneration is a conserved function of TIR-1 by asking whether motor neuron-specific expression of human SARM1 is also capable of inhibiting axon regeneration. As we observed with *tir-1*, expressing an active form of human *Sarm1* containing its SAM and TIR domains (hSarm1$_{SAMTIR}$) in GABA motor neurons was sufficient to rescue the regeneration phenotype of *tir-1(qd4)* mutants (*Figure 2F*). Together, these results demonstrate that the ability to inhibit axon regeneration cell-autonomously is conserved in the human SARM1$_{SAMTIR}$ protein.

## A novel model of injury-induced motor axon degeneration in *C. elegans*

The finding that TIR-1 and human SARM1 can function intrinsically to inhibit axon regeneration is unexpected since *Sarm1* and *dSarm* promote the opposite response to injury, axon degeneration, in mice and flies (*Gerdts et al., 2013*; *Osterloh et al., 2012*). Whether TIR-1 also promotes injury-induced motor axon degeneration in *C. elegans* is not known. To answer that question, we needed to examine the consequences of mutating *tir-1* in an injury paradigm that induces significant axon degeneration. Although the above axotomy assay, in which an axon is severed once, is extremely useful for investigating the genetic and cellular mechanisms that regulate axon regeneration (*Byrne and Hammarlund, 2017*), it is not as robust a model of injury-induced axon degeneration. The obstacle is that while proximal segments, which remain attached to the cell body, regenerate in the 24 hr following injury (*Yanik et al., 2004*), the distal segments, which are no longer attached to the cell body, remain largely intact (*Nichols et al., 2016*). This prompted us to develop an assay with which robust injury-induced motor axon degeneration and axon regeneration can be investigated simultaneously in vivo. Here,

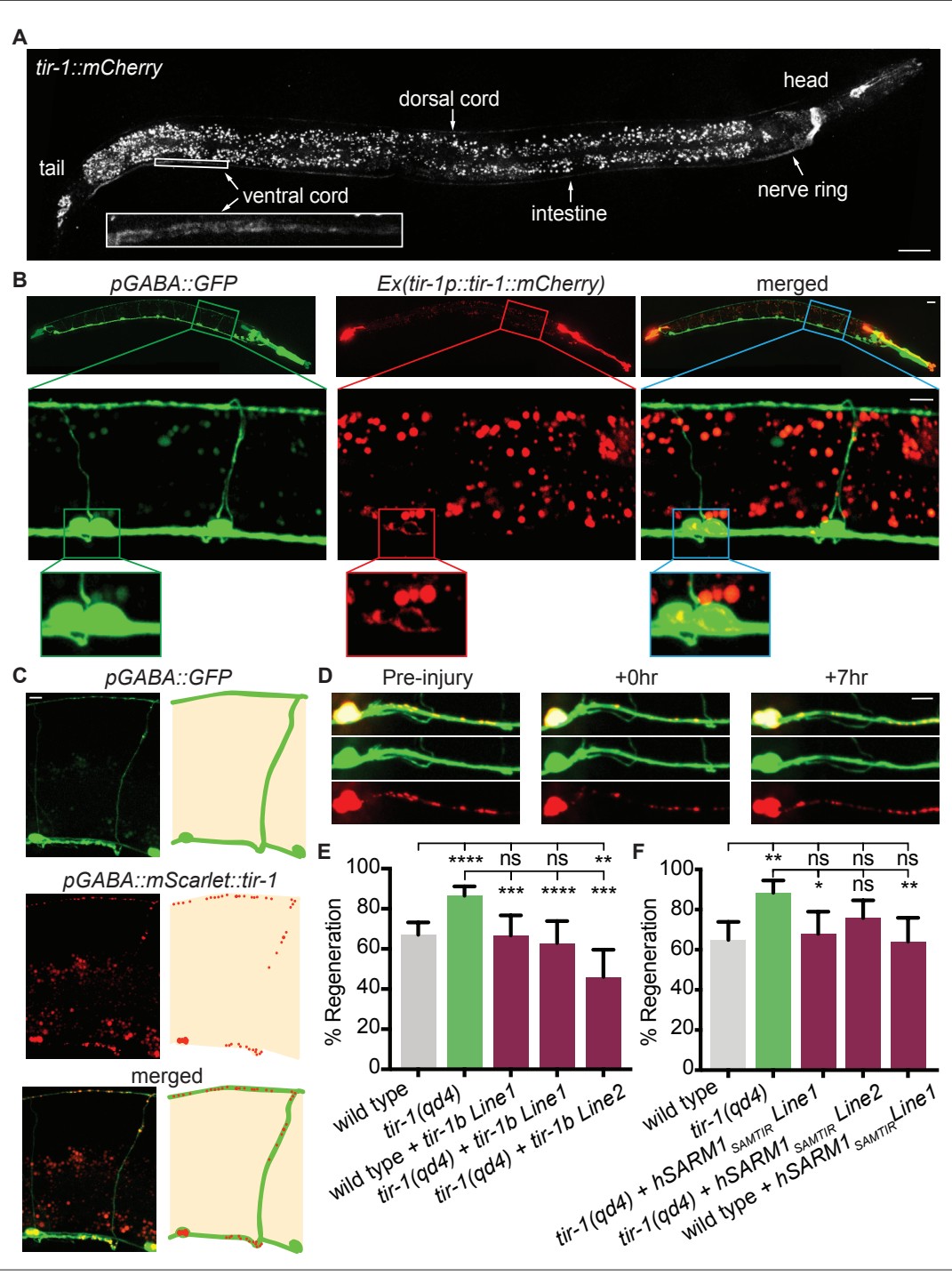

**Figure 2.** TIR-1 functions cell-autonomously to inhibit regeneration. (**A**) Endogenous TIR-1 is expressed faintly in the nervous system, particularly in neurons along the dorsal and ventral (enlarged box) nerve cords. Note the intestinal expression significantly overlaps with autofluorescent gut granules and therefore is not an accurate representation of endogenous *tir-1* expression in the intestine. Scale bar:25 µm. (**B**) Representative micrographs of transgenic animals expressing *tir-1b::mCherry* under its endogenous promoter indicates TIR-1b::mCherry is expressed in GABA neurons. Enlarged boxes show TIR-1b::mCherry expression in the cell bodies of GABA neurons. Scale bars: upper images, 25 µm; insets, 10 µm. (**C,D**) When expressed specifically in GABA neurons, *mScarlet::tir-1b* is expressed in a punctate pattern in the cell bodies and axons of GABA neurons before, immediately after and 7 hr after injury. Scale bars: 5 µm. (**E**) GABA-specific expression of *tir-1b* rescues axon regeneration in *tir-1(qd4)* animals but not wild-type animals. N = 197, 148, 62, 50, 69. (**F**) GABA-specific expression

*Figure 2 continued on next page*

*Figure 2 continued*

of human SARM1$_{SAM-TIR}$ lacking the N-terminal autoinhibitory domain rescues axon regeneration in single cut *tir-1(qd4)* mutants and not in wild-type animals. N = 91, 60, 53, 66, 50. Significance relative to wild type or *tir-1(qd4)* is indicated by *p≤0.05, **p≤0.01, ***p≤0.001, Fisher's exact test. Error bars represent 95% confidence intervals.

rather than severing axons once, we severed individual axons in two places, approximately 25 μm apart, which creates a middle fragment that degenerates (*Figure 3A*). The degeneration has similar morphological and temporal features as Wallerian degeneration, where after a brief latent period, the middle axon segment beads, fragments, and is eventually cleared (*Figure 3B*). We tested how reliably double injury induces degeneration of the middle fragment by quantifying the percentage of severed axons whose middle fragment either degenerated (complete clearance), partially degenerated (>65% of middle fragment degenerated) or largely remained intact (>80% of middle fragment remained). In wild-type axons, 90.2% of middle fragments degenerated 24 hr after double axotomy (*Figure 3C*). We did not observe a significant difference in the number of middle fragments when visualized with either cytosolic or myristoylated and membrane-bound GFP 1.5 hr and 24 hr after injury (*Figure 3D*), indicating that the observed degeneration is not caused by a release of cytosolic GFP from the severed fragments. Notably, significantly less axons degenerated 1.5 hr after injury compared to 24 hr after injury, suggesting the earlier time point represents a period of active degeneration (*Figure 3E*). Finally, we found that injured motor axons are less likely to degenerate in aged animals, indicating an active destruction process is either lost or suppressed with increased age (*Figure 3F*). Together, these data indicate that the observed degeneration is an active process, not caused by a leaky cytosolic reporter and that the double cut assay reliably induces axon degeneration that can be investigated in vivo and with single-neuron precision.

## *tir-1* is a functional ortholog of *Sarm1* that promotes injury-induced axon degeneration

*Sarm1/dSarm* is a key regulator of injury-induced degeneration in multiple species, including mammals, flies, and zebrafish (*Gerdts et al., 2013*; *Osterloh et al., 2012*; *Tian et al., 2020*). To determine whether *tir-1* is a functional ortholog of *Sarm1/dSarm* with a conserved role in injury-induced axon degeneration, we assayed its ability to regulate degeneration using the double-cut assay. We found that in *tir-1* mutant animals, only 68% of middle fragments degenerated compared to 86% of middle fragments in wild-type animals (*Figure 3G*). Therefore, like *Sarm1/dSarm*, *tir-1* positively regulates injury-induced axon degeneration. Expression of *tir-1b* specifically in GABA motor neurons restored middle fragment degeneration in *tir-1(qd4)* animals, indicating *tir-1* functions cell-autonomously to regulate injury-induced axon degeneration (*Figure 3H*).

Since loss of *tir-1* incompletely suppressed axon degeneration, we investigated whether *ced-1*, a homolog of *Draper* and *Megf10*, also contributes to the degeneration phenotype. CED-1 functions in the hypodermis and muscle cells of *C. elegans* to remove axonal debris after injury in mechanosensory neurons (*Chiu et al., 2018*; *Nichols et al., 2016*). We found that injured motor axons in *ced-1* loss-of-function animals degenerated as frequently as wild-type axons after double axotomy (*Figure 3—figure supplement 1*). Moreover, animals with loss-of-function mutations in both *ced-1* and *tir-1* phenocopied the amount of degeneration observed in *tir-1* animals. These results demonstrate that middle fragment degeneration after double injury does not depend on *ced-1* and that *tir-1* functions independently from *ced-1* to regulate injury-induced degeneration. Together with our finding that injury-induced degeneration is not completely dependent on *tir-1* (*Figure 3F*), these data indicate *C. elegans* motor axons have evolved additional as yet unidentified regulators of injury-induced motor axon degeneration.

We hypothesized that perhaps distal fragments do not degenerate after single axotomy because TIR-1 is not sufficiently activated by this type of injury. TIR-1 and SARM1 functions are autoinhibited by intramolecular and intermolecular interactions with their N-terminal domains (*Chuang and Bargmann, 2005*; *Figley et al., 2021*; *Gerdts et al., 2013*; *Horsefield et al., 2019*; *Jiang et al., 2020*; *Shen et al., 2021*; *Sporny et al., 2020*; *Summers et al., 2016*). We asked whether low-level expression of an endogenous isoform lacking the autoinhibitory domain was sufficient to induce degeneration of GFP-labeled distal fragments, even after a single injury (*Figure 4*). Strikingly, we found that low levels of GABA-specific TIR-1b expression caused the majority of distal stumps to degenerate after

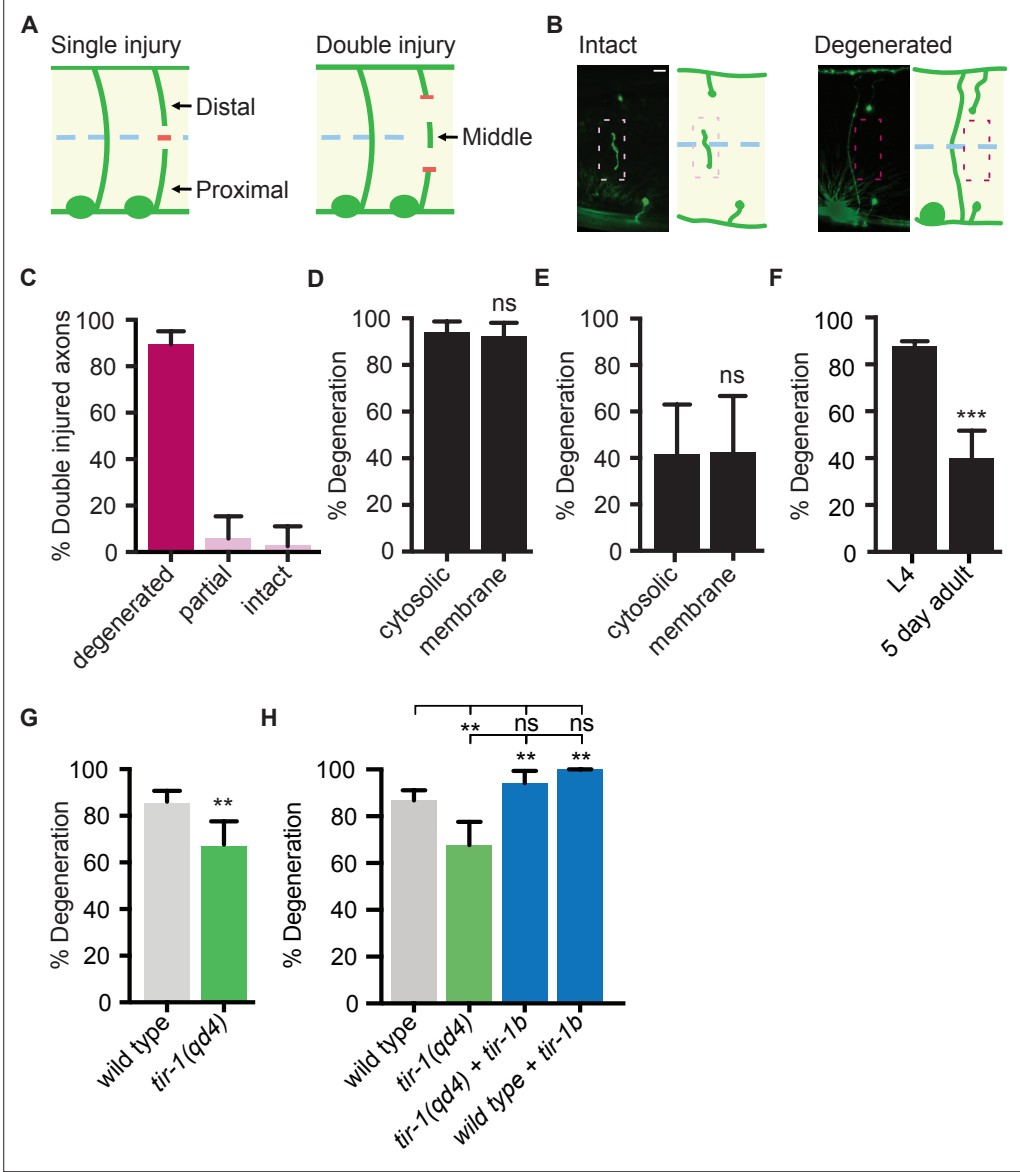

**Figure 3.** TIR-1 promotes axon degeneration in injured GABA motor neurons. (**A**) Single and double injury models. After being severed once, GABA axons do not degenerate. Severing an axon twice creates a middle fragment that degenerates. (**B**) Micrographs of intact and degenerated middle fragments 24 hr post axotomy. Boxes with dashed lines highlight intact (pink) and degenerated (red) middle fragments. Scale bar: 5 μm. (**C**) Most middle fragments degenerate 24 hr after double injury. N = 61. (**D**) No significant difference in middle fragment degeneration was observed between axons expressing membrane tagged- or cytosolic GFP 24 hr (N = 19, 14) and (**E**) 1.5 hr after injury. (**F**) Middle fragments degenerate less frequently in aged adult animals compared to animals in the fourth larval stage (L4). N = 70, 44. (**G**) *tir-1* promotes middle fragment degeneration. N = 63, 68. (**H**) Expression of *tir-1b* specifically in GABA motor neurons restores regeneration to *tir-1(qd4)* animals. N = 211, 68, 34, 37. Significance relative to control is indicated by *p≤0.05, **p≤0.01, ***p≤0.001, Fisher's exact test. Error bars represent 95% confidence intervals.

The online version of this article includes the following figure supplement(s) for figure 3:

**Figure supplement 1.** CED-1 does not regulate middle fragment degeneration.

a single axotomy (*Figure 4A–E*). Notably, degeneration was not observed in the proximal stumps, indicating that despite the lack of its autoinhibitory domain, TIR-1 specifically regulates degeneration distal to the site of injury. Therefore, in wild-type *C. elegans*, axons that have been injured once are likely incapable of degenerating, at least in part, because the injury response was of insufficient

magnitude or identity to activate TIR-1. In contrast, a double axotomy may induce an enhanced injury response in the middle fragment, which is of sufficient magnitude or identity to activate TIR-1 or overcome its suppression.

To further investigate the extent to which TIR-1 is able to promote axon degeneration, we asked whether overexpressing TIR-1 is sufficient to induce motor axon degeneration in the absence of injury. When we expressed *tir-1b* at high concentrations, we found that GFP-labeled motor axons were beaded, severely fragmented, and degenerated even in the absence of injury (*Figure 4F–H*). As a result, there was a significant decrease in the amount of GFP along the axon commissures and dorsal nerve cord of *tir-1b* overexpressing animals compared to the amount of GFP in wild-type controls (*Figure 4I and J*). The most obvious phenotype was the virtual absence of axon commissures that extended fully from the ventral to the dorsal nerve cord in *tir-1* transgenic animals compared to wild-type animals (*Figure 4K*). Despite the absence of axons, the number of GABA neuron cell bodies was consistent between transgenic and non-transgenic animals, indicating *tir-1* overexpression selectively induces axon degeneration and not cell death (*Figure 4G*). Importantly, the *tir-1* transgene was expressed as an extrachromosomal array, resulting in mosaic GABA motor neuron expression. This mosaicism allowed us to find that the few non-transgenic GABA motor axons had a wild-type morphology, indicating the observed degeneration was caused by cell-autonomous overexpression of the *tir-1b* transgene and was not merely a secondary consequence of physically manipulating and imaging the animals (*Figure 4—figure supplement 1*). Therefore, elevated *tir-1b* expression within GABA neurons is sufficient to induce severe and chronic axon degeneration, even in the absence of injury. Together with the finding that loss of *tir-1* function suppresses injury-induced axon degeneration, these data demonstrate that *tir-1* positively regulates axon degeneration when it is constitutively activated or sufficiently activated by injury.

## TIR-1 regulates axon regeneration independently of its NADase activity and with the NSY-1–PMK-1 mitogen-activated protein kinase signaling cascade

How does one gene regulate both seemingly opposite processes of repair and destruction in what was once the same cell? We proceeded to investigate how TIR-1 regulates both regeneration and degeneration using the double injury model. The ability to observe both axon regeneration and degeneration using the same injury model is essential for comparing how TIR-1 regulates the two seemingly opposite processes (1) within the same injured wild-type axon and (2) in the context of the same injury response. Indeed, the importance of comparing aspects of the injury response within the same injury model is manifest in the finding that axons regenerate significantly less frequently after a double injury compared to a single injury (60.53% of axons regenerate in the 24 hr following a single injury, and 37.38% of axons regenerate in the 24 hr following a double injury). For these reasons, unless otherwise indicated, the experiments below were performed using double axotomy.

SARM1 and dSarm execute several cellular functions in injured axons, including cleavage of NAD$^+$, regulation of MAPK signaling and the Axundead protein, as well as regulation of calcium signaling, gene transcription, and autophagy (*Coleman and Höke, 2020*; *Essuman et al., 2017*; *Neukomm et al., 2017*; *Walker et al., 2017*; *Yang et al., 2015*). Early investigations of TIR-1 function revealed that it also signals through variations of a canonical downstream MAPK pathway in the innate immune response and in neuronal development (*Chuang and Bargmann, 2005*; *Couillault et al., 2004*; *Liberati et al., 2004*; *Pujol et al., 2008*; *Sagasti et al., 2001*). The molecular mechanisms that mediate TIR-1, SARM1, and dSarm functions informed a candidate approach to dissect the genetic pathways in which TIR-1 regulates both axon regeneration and degeneration of injured axons.

The NAD$^+$ hydrolase activity of SARM1 is conserved in the *C. elegans* TIR domain, in a TIR-1 isoform lacking the N-terminal inhibitory domain, and in the full-length TIR-1 protein (*Gerdts et al., 2016*; *Horsefield et al., 2019*; *Loring et al., 2021*; *Loring et al., 2020*; *Peterson et al., 2022*; *Summers et al., 2016*). In SARM1, TIR domain function requires an active site containing the key catalytic residue E642, a structure specific to catalytically active TIR domains called a SARM-specific (SS) loop (residues 622–635), a BB loop (residues 594–605) thought to be important for intradomain interactions, and multiple residues important for interaction with the autoinhibitory domain and protein multimerization (*Essuman et al., 2017*; *Horsefield et al., 2019*; *Loring et al., 2021*; *Loring et al., 2020*; *Summers et al., 2016*). To directly address whether TIR-1 regulates regeneration by

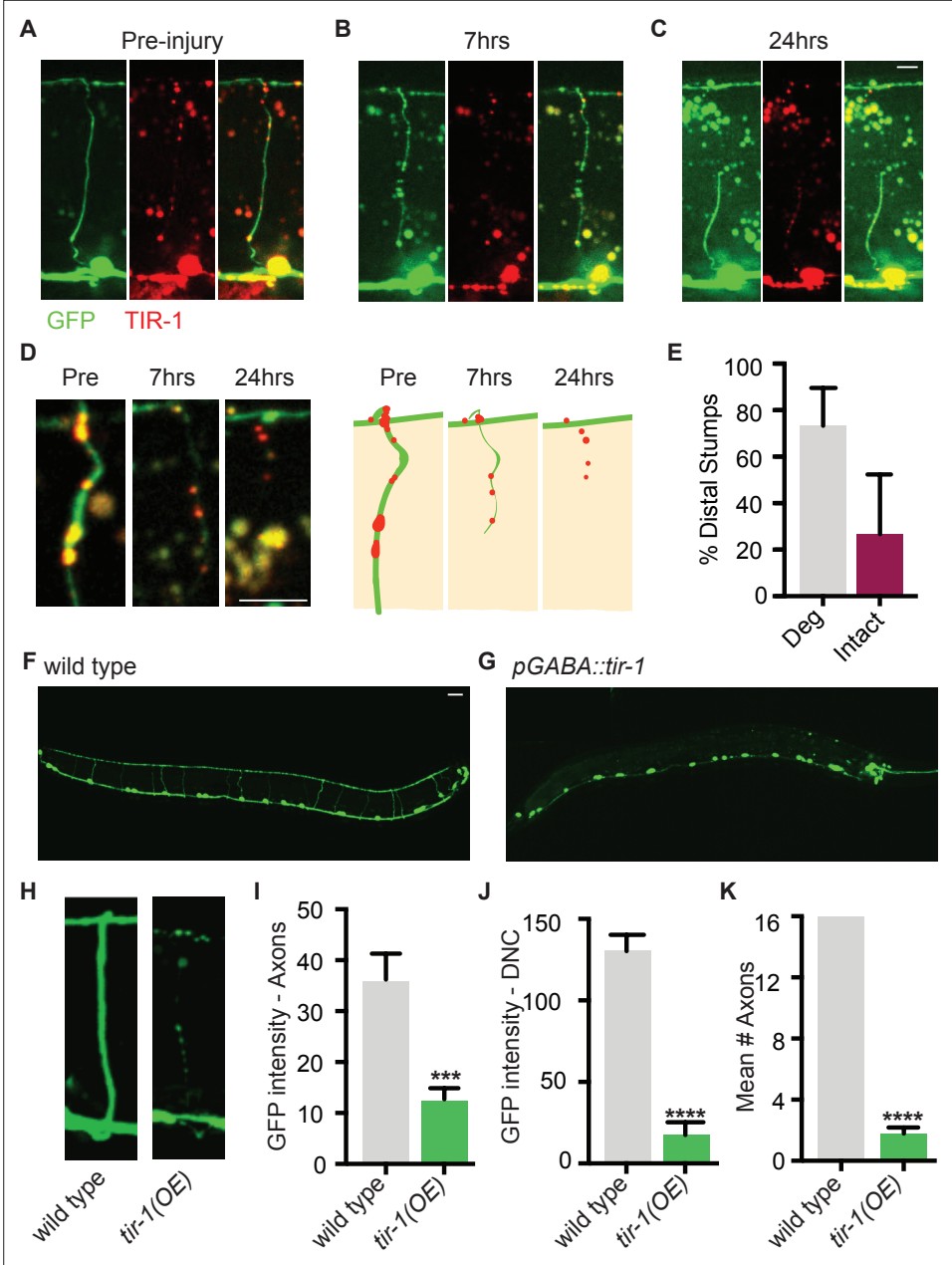

**Figure 4.** TIR-1 expression causes spontaneous axon degeneration in GABA motor neurons. (**A–C**) Expression of *mScarlet::tir-1b* in GFP-labeled GABA motor neurons is sufficient to induce degeneration of the distal stump after a single axotomy. Micrographs and representative drawings demonstrate mScarlet::TIR-1b is localized in the cytoplasm in GABA cell bodies and along their axons in the ventral nerve cord, commissures, and dorsal nerve cord. Three timepoints are represented: (**A**) before injury, (**B**) immediately after injury, and (**C**) 7 hr after injury. At 7 hr, the distal stump has degenerated while the proximal stump remains intact. Scale bar: 5 µm. (**D**) Magnified image of a degenerating distal stump before injury, 7 hr after injury and 24 hr after injury. Scale bar: 5 µm. (**E**) Most distal stumps degenerate within 24 hr after a single injury in animals expressing *mScarlet::tir-1b*. (**F, G**) Individual GABA motor axons (seen in **F**) degenerate in transgenic animals expressing *unc-47p::tir-1b::mCherry* (**G**). Scale bar: 20 µm. (**H**) Magnified image of a wild-type axon and one that degenerates in animals that express *unc-47p::tir-1b::mCherry.* There is less (**I**) GFP intensity along the axon commissures (N = 18, 14), (**J**) GFP intensity along the dorsal nerve cord and (**K**) number of intact axons in animals that overexpress *tir-1* in GABA neurons compared to wild-type animals (N = 6, 9), indicating *tir-1* promotes axon degeneration. For categorical data, significance relative to wild type is indicated by ****p≤0.0001, Fisher's exact test. Error bars represent 95% confidence intervals. For

*Figure 4 continued on next page*

*Figure 4 continued*

continuous data, significance relative to wild type is indicated by ***p≤0.001, Student's *t*-test. Error bars represent SEM.

The online version of this article includes the following figure supplement(s) for figure 4:

**Figure supplement 1.** TIR-1 expression induces degeneration.

hydrolyzing NAD⁺, we CRISPR edited the endogenous *tir-1* gene to create mutations in the key catalytic residue (*tir-1(E788A)*) and in the SS loop (*tir-1(D773A)*), which correspond with SARM1$^{E642A}$ and SARM1$^{D627A}$ mutants, respectively (*Essuman et al., 2017*; *Loring et al., 2021*; *Summers et al., 2016*). The conserved E788/E642/E1170 catalytic residue is required for the NADase activity of the TIR-1, SARM1, and dSarm TIR domains (*Brace et al., 2022*; *Essuman et al., 2017*; *Herrmann et al., 2022*; *Horsefield et al., 2019*; *Hsu et al., 2021*; *Loring et al., 2021*; *Peterson et al., 2022*; *Summers et al., 2016*). The D773/D627 aspartate is also conserved in the SS loops of *C. elegans* and human TIR domains. It is required for NAD⁺ catalysis by SARM1 and its ability to induce degeneration of injured DRGs but not for its ability to multimerize (*Loring et al., 2021*; *Summers et al., 2016*). We found that both *tir-1(E788A)* and *tir-1(D773A)* mutant animals displayed wild-type levels of axon regeneration, indicating that TIR-1$^{E788A}$ and TIR-1$^{D773A}$ are functionally expressed and that neither residue are required for TIR-1 to inhibit axon regeneration (*Figure 5A*). Therefore, TIR-1 regulates regeneration independently of its NADase activity.

We began our search for NADase-independent mechanisms of TIR-1 function by investigating whether TIR-1 inhibits regeneration with established regulators of the injury response. Perhaps the most deterministic regulator of axon regeneration in *C. elegans* motor neurons is the mitogen-activated protein kinase kinase kinase DLK-1 (Dual leucine zipper kinase-1). DLK-1 function is absolutely required for the regeneration of otherwise wild-type GABA motor axons (*Hammarlund et al., 2009*; *Yan et al., 2009*). We found that loss of *tir-1* function did not suppress the complete absence of regeneration in either the *dlk-1* loss-of-function mutant or the downstream *pmk-3/*p38 MAPK loss-of-function mutant, indicating *tir-1* either functions upstream of *dlk-1* signaling or in a separate signaling pathway (*Figure 5B*). Physical interpretations of genetic interactions with genes that are essential for a given function can be difficult if that function requires the activity of multiple distinctly regulated processes. Therefore, since DLK-1 function is absolutely required for axon regeneration, these interactions do not distinguish between the possibilities that TIR-1 signaling regulates the activity of DLK-1 or that DLK-1 regulates an essential component of axon regeneration independently of TIR-1 signaling. However, the genetic interactions do indicate that TIR-1 does not function downstream of, or redundantly with, DLK-1 to regulate axon regeneration.

We next asked whether previously identified TIR-1 interactors mediate its role in axon regeneration. Initial studies of TIR-1 identified a shared MAPK signaling cassette that interacts with TIR-1 to regulate neuronal development and innate immunity (*Chuang and Bargmann, 2005*; *Couillault et al., 2004*; *Liberati et al., 2004*; *Pujol et al., 2008*; *Sagasti et al., 2001*). During neuronal development, TIR-1 interacts with the calcium-calmodulin-dependent protein kinase II (CaMKII) ortholog UNC-43 and the MAP kinase kinase kinase NSY-1/ASK1 signaling pathway to specify the asymmetric neuronal identity of AWC olfactory neurons. Specifically, in response to calcium, TIR-1 and UNC-43 form a postsynaptic protein complex that transiently interacts with NSY-1, which in turn regulates odorant receptor expression through its downstream MAP kinases (*Chuang and Bargmann, 2005*; *Sagasti et al., 2001*). In contrast, the TIR-1/NSY-1 signaling cassette functions independently of UNC-43 to regulate the innate immune response (*Couillault et al., 2004*; *Liberati et al., 2004*; *Pujol et al., 2008*). We first asked whether TIR-1 regulates axon regeneration by interacting with the NSY-1 signal transduction pathway and found that loss of *nsy-1* or its downstream MAP kinase *pmk-1/*p38 enhanced axon regeneration after double injury (*Figure 5C and F*). These regeneration phenotypes were not significantly enhanced by deletion of *tir-1*. Because *tir-1*, *nsy-1*, and *pmk-1* null mutants all inhibit axon regeneration to the same extent, and because the amount of regeneration in any pairwise combination of mutants is not additive or multiplicative compared to the single mutants, our data indicate that the three genes function in the same pathway to regulate axon regeneration after double injury (*Figure 5F*). Continuing our investigation of candidate interactors, we considered *unc-43* because it was previously found to inhibit regeneration of sensory axons (*Chung et al., 2016*). We found that *unc-43(e408)* phenocopied the increased axon regeneration of *tir-1(qd4)* mutants and addition of the *unc-43(e408)* mutation to

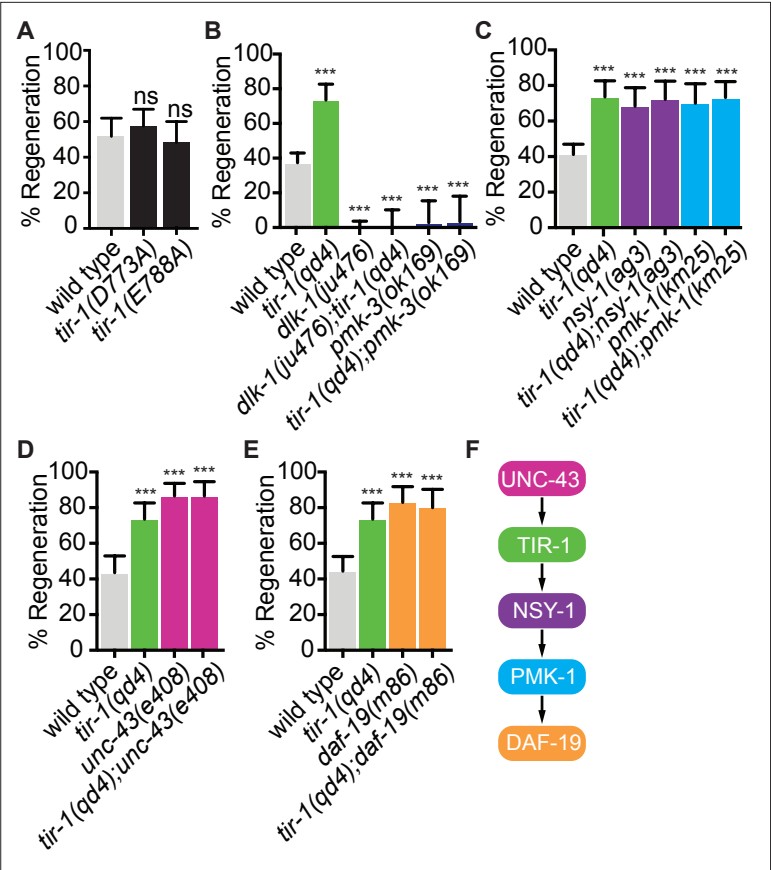

**Figure 5.** TIR-1 functions independently of its NADase activity and with the NSY-1 signaling pathway to inhibit axon regeneration. (**A**) Conserved residues E788A/E642A, which is required for the NADase function of TIR-1 and SARM1, and D773A/D627 are not required to inhibit axon regeneration. (**B**) DLK-1 and TIR-1 function in the same or parallel pathways to regulate axon regeneration. N = 288, 68, 122, 41, 36, 30. (**C**) Axon regeneration is significantly increased in predicted null alleles of *nsy-1/ASK1* and *pmk-1/p38* following double injury, in both the presence and absence of *tir-1* function. Double mutants are not statistically different from single mutants. N = 289, 68, 60, 50, 54, 71. (**D**) Null mutation of *unc-43/CAMKII* enhances axon regeneration after double injury in the presence and absence of *tir-1*. N = 89, 68, 39, 37. (**E**) Loss of the transcription factor *daf-19/RFX1-3* increases axon regeneration after double injury in the presence and absence of *tir-1*. N = 142, 68, 41, 35. (**F**) Components of the TIR-1–NSY-1 signaling cascade. Significance relative to wild type is indicated by *p≤0.05, **p≤0.01, ***p≤0.001. Error bars represent 95% confidence intervals.

The online version of this article includes the following figure supplement(s) for figure 5:

**Figure supplement 1.** TIR-1 inhibits regeneration with the NSY-1/ASK1 and PMK-1/p38 MAPK after a single injury.

**Figure supplement 2.** The ATF-7 transcription factor does not regulate axon regeneration.

*tir-1(qd4)* mutants did not further enhance regeneration (***Figure 5D***). Therefore, *tir-1 and unc-43* function together in a shared genetic pathway to inhibit axon regeneration after double injury. Since the *nsy-1* and *pmk-1* null phenotypes fully recapitulate the *tir-1(qd4)* phenotype and because they have the opposite phenotype to *dlk-1* mutants, the *nsy-1* and *dlk-1* MAP kinase pathways do not function redundantly to inhibit axon regeneration downstream of *tir-1*. Together, these results indicate that *tir-1* and the *nsy-1–pmk-1* MAPK pathway function together to inhibit axon regeneration, which is dependent on *dlk-1*.

After a single injury, the regeneration frequency and genetic interactions between *tir-1*, *nsy-1*, and *pmk-1* null mutants were similar to those after a double injury and wild-type regeneration was not affected by the lack of NADase activity in *tir-1(E788A)* animals (***Figure 5—figure supplement 1***). Interestingly, although wild-type axons regenerate less frequently after a double injury compared to a single injury, the magnitude of regeneration in TIR-1, NSY-1, and PMK-1 mutants is similar in axons

that have been injured once (*Figure 5—figure supplement 1A and B*) or twice (*Figure 5C*). Therefore, the TIR-1–NSY-1–PMK-1 pathway not only inhibits regeneration after both types of injury, it may be epistatic to what differentiates the regeneration frequency between the two types of injury.

## TIR-1 inhibits axon regeneration by regulating the RFX-type transcription factor DAF-19

CaMKII has previously been found to regulate axon regeneration and developmental axon outgrowth, in part by regulating transcription of effector genes (*Chung et al., 2016*; *Ghiretti et al., 2014*; *Xi et al., 2019*). However, the signaling pathways and set of transcription factors that mediate the effects of CaMKII activation on target gene expression during regeneration are not known. Given the genetic interactions between *tir-1*, *unc-43*, *nsy-1*, and *pmk-1*, we hypothesized that UNC-43 might regulate axon regeneration via TIR-1 and NSY-1-dependent transcription. We examined this possibility by testing the regenerative ability of animals with predicted null mutations in the cAMP-dependent transcription factor *atf-7/ATF2/CREB5* and in the Regulatory Factor X transcription factor *daf-19/RFX*, both of which are regulated by *nsy-1* signaling (*Shivers et al., 2009*; *Vériépe et al., 2015*; *Xie et al., 2013*). Loss of *atf-7* function did not affect GABA axon regeneration, consistent with previous studies of PLM mechanosensory axon regeneration (*Figure 5—figure supplement 2*; *Ghosh-Roy et al., 2010*). However, loss of *daf-19* function increased regeneration to the same extent as loss of *tir-1* and did not enhance the regeneration phenotype of the *tir-1* null mutants (*Figure 5D*). Together, these data are consistent with a model in which an *unc-43/tir-1/nsy-1/pmk-1/daf-19* signaling cascade (*Figure 5E*) inhibits axon regeneration by altering transcription of downstream regeneration-associated genes.

## TIR-1 regulates degeneration in coordination with DLK-1 MAPK signaling

To determine how TIR-1 achieves specificity of function on either side of the injury, we also asked how it promotes axon degeneration. Since SARM1 and TIR-1 are established NAD$^+$ hydrolases (*Essuman et al., 2017*; *Horsefield et al., 2019*; *Loring et al., 2021*; *Loring et al., 2020*; *Peterson et al., 2022*; *Summers et al., 2016*), we began by asking whether TIR-1 regulates regeneration by hydrolyzing NAD$^+$ in injured axons. To do so, we quantified degeneration in TIR-1$^{E788A}$ animals (homologous to SARM1$^{E642A}$), which renders the TIR domain of *C. elegans* TIR-1 catalytically dead (*Essuman et al., 2017*; *Loring et al., 2020*; *Peterson et al., 2022*). Interestingly, the E788A/E642A mutation did not suppress injury-induced axon degeneration (*Figure 6A*), indicating TIR-1 can regulate axon degeneration independently of its NAD$^+$ hydrolase activity. This ability contrasts sharply with the requirement for NAD$^+$ hydrolysis in dSarm and SARM1-mediated degeneration (*Brace et al., 2022*; *Essuman et al., 2017*; *Gerdts et al., 2015*; *Herrmann et al., 2022*; *Hsu et al., 2021*), indicating TIR-1 regulates a previously unexplored mechanism of axon degeneration.

We then asked whether other components of the TIR domain are required for its ability to regulate axon degeneration. Specifically, we investigated residue D773, which is homologous to D627 in the SS-loop of SARM1 that is required for injury-induced degeneration of DRG axons yet promotes the TIR domain's NADase activity through unspecified mechanisms (*Loring et al., 2020*; *Summers et al., 2016*). We found that axon degeneration in *tir-1(D773A)* mutants was slightly but significantly disrupted compared to wild-type animals and was similar to that seen in *tir-1(qd4)* animals (*Figure 6A*). These data indicate the SS loop is important for TIR-1's ability to promote degeneration of injured axons. Collectively, the findings that mutation of the key catalytic glutamate E788A abolishes NADase activity (*Peterson et al., 2022*) but not degeneration, along with previous data showing the SS-loop residue D627 promotes the NADase activity of SARM1 through relatively unknown mechanisms (*Loring et al., 2020*; *Summers et al., 2016*), indicate that the D773A/D627A mutation suppresses degeneration by disrupting an as yet unidentified function of TIR-1. Notably, D773A/D627A is not a null mutation since it did not disrupt the ability of TIR-1 to inhibit regeneration (*Figure 5A*). It follows that TIR-1 regulates axon regeneration and degeneration by different mechanisms, potentially by directly interacting with components of the UNC-43-NSY-1-PMK-1 pathway on the proximal side of the injury and another signal transduction pathway on the distal side of the injury.

The surprising finding that TIR-1 is capable of NADase-independent degeneration in response to injury prompted us to ask whether TIR-1 is capable of inducing NADase-dependent axon degeneration in other contexts. We found that while expressing large amounts of wild-type *tir-1b* induced

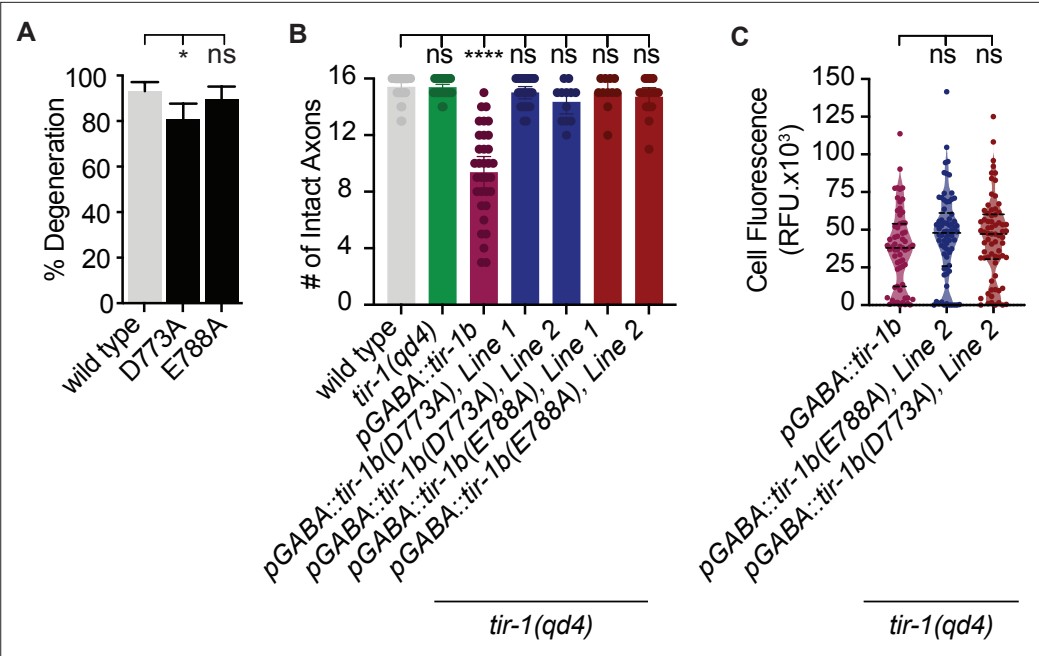

**Figure 6.** TIR-1 is capable of NADase-independent and -dependent axon degeneration. (**A**) Mutation of conserved residue E788A/E642A, which is required for the NADase function of TIR-1 and SARM1, does not affect axon degeneration after double axotomy. D773A/D627A modestly suppresses degeneration. N = 97, 106, 64. (**B**) Conserved residues E788A/E642A and D773A/D627A are required for chronic axon degeneration. Lines represent independently generated transgenic strains that were generated by injecting equal concentrations of the same *tir-1b*::mCherry expression plasmid into separate *tir-1(qd4)* animals and isolating transgenic lines from each animal. N = 34, 32, 33, 23, 1211, 17. (**C**) Equivalent amounts of wild-type *tir-1b*, *tir-1b(e788a)*, and *tir-b(d773a)* transgenes are expressed, as quantified by total mCherry fluorescence in the cell bodies of *tir-1(qd4)* animals, which were identified by expression of an integrated GABA-specific GFP marker, *oxIs12* (N = 63, 70, 70). Significance relative to wild type was determined by one-way ANOVA and Dunnett's test and is indicated by ****p≤0.0001. Error bars represent 95% confidence intervals.

significant axon degeneration even in the absence of injury, expressing either *tir-1b(E788A/E642A)* or *tir-1b(D773A/D627A)* did not (***Figure 6B***). The absence of spontaneous degeneration in the presence of TIR-1b(D773A) and TIR-1(E788A) is not likely attributed to differential expression of the mutant and wild-type transgenes, which was determined by quantifying expression of the fluorescent tag on each protein (***Figure 6C***). However, because a small but statistically insignificant number of damaged axons were observed, we cannot rule out the possibility that greater TIR-1b(D773A) or TIR-1b(E788A) expression might induce NADase-independent degeneration. Together with our previous results, we find that D773 is required for both injury-induced and spontaneous degeneration, while E788 is only required for spontaneous degeneration. These data indicate D773A disrupts multiple facets of TIR-1 function, and the NADase activity of TIR-1 is required to induce spontaneous degeneration when overexpressed in uninjured axons but is not required for degeneration of injured axon fragments.

Therefore, the question remains: how does TIR-1 regulate NADase-independent degeneration of injured axons? We first investigated whether TIR-1 signals with the same *nsy-1–pmk-1* MAPK pathway that mediates its function on the proximal side of the injury by measuring the presence or absence of middle fragments 24 hr after double injury in the absence of the MAPK *pmk-1*. Loss of *pmk-1* did not have a degeneration phenotype on its own and double *tir-1; pmk-1* mutant animals phenocopied the degeneration observed in *tir-1* mutant animals (***Figure 7A***). This lack of genetic interaction indicates that *tir-1* functions independently from *pmk-1* to promote axon degeneration after injury.

DLK MAPK-signaling functions with SARM1/dSarm to regulate axon degeneration in mice and flies (***Miller et al., 2009***; ***Walker et al., 2017***; ***Welsbie et al., 2013***; ***Xiong and Collins, 2012***; ***Xiong et al., 2010***; ***Yang et al., 2015***). In our assay, loss of either *dlk-1* or its downstream p38 MAP kinase *pmk-3* did not affect axon degeneration on their own; however, they did rescue axon degeneration in *tir-1* mutants (***Figure 7B***). This result suggests the decreased axon degeneration phenotype of

*tir-1* mutants depends on *dlk-1* pathway activity. However, due to the high amount of axon degeneration in wild-type animals, it is possible that an upper threshold in the assay prevents us from seeing enhanced degeneration above wild-type levels. Therefore, to further investigate whether *dlk-1* inhibits axon degeneration, we asked whether overexpression of *dlk-1* would protect axons. We found that, indeed, significantly less severed axon fragments degenerated in *dlk-1(oe)* animals relative to wild-type animals (*Figure 7B and C*, *Figure 7—figure supplement 1*). Of note, DLK-1 function is not responsible for inhibiting degeneration after a single injury as neither loss or gain of *dlk-1* function induced degeneration of singly injured distal axon fragments (*Figure 7—figure supplement 2*). Therefore, DLK-1's ability to inhibit degeneration is context specific.

The suppression of degeneration in doubly injured *dlk-1(oe)* axons was not dependent on *tir-1* function (*Figure 7B*). However, loss of *pmk-3*/p38 MAPK function rescued degeneration in both *dlk-1(oe)* and *tir-1(-)* animals, as well as in double *dlk-1(oe); tir-1(-)* mutants (*Figure 7B*). The simplest interpretation of these genetic epistasis analyses is that PMK-3 inhibits degeneration downstream of its canonical MAPKKK DLK-1, and also functions downstream of TIR-1. These findings support a model in which TIR-1 promotes degeneration by antagonizing or opposing the inhibitory role of DLK-1-PMK-3 signaling. However, the exact nature of the complex relationship between TIR-1, DLK-1, and PMK-3 are exciting future avenues of investigation, as is the question of how degeneration is effected in the absence of TIR-1. Together, our data indicate that TIR-1 promotes axon degeneration by directly or indirectly antagonizing DLK-1-PMK-3 function on the distal side of the injury and inhibits regeneration with UNC-43–NSY-1–PMK-1–DAF-19 signaling on the proximal side of the injury (*Figure 7D*).

## Discussion

Understanding the mechanisms that determine whether axons regenerate and degenerate is critical to our ability to stimulate repair after neuronal injury (*Girouard et al., 2018*). Here, we identify TIR-1/SARM1/dSarm, a central regulator of axon degeneration, as a previously uncharacterized intrinsic inhibitor of axon regeneration that can be targeted to increase repair of injured motor axons. Our data indicate that to achieve this specificity of function, *tir-1* inhibits axon regeneration with the NSY-1–PMK-1 MAPK pathway on the proximal side of the injury while it promotes degeneration by antagonizing the DLK-1–PMK-3 MAPK pathway on the distal side of the injury. Together, our findings reveal divergent mechanisms by which TIR-1 regulates the injury response.

### TIR-1/SARM1 functions intrinsically to regulate both axon regeneration and axon degeneration

We found that TIR-1 inhibits axon regeneration and does so independently of its role in degeneration by investigating the injury response in a new model of *C. elegans* motor axon injury. *C. elegans* has been a powerful model of axon regeneration owing to its genetic tractability, the ability to carry out experiments in vivo with single-axon resolution, and because the majority of its genes are conserved with mammals (*Byrne et al., 2011*; *Kim et al., 2018b*; *Yanik et al., 2004*). In addition, the *C. elegans* motor nervous system lacks myelin and glia, enabling investigation of mechanisms that regulate axon regeneration independently of these extrinsic cues. The finding that both *tir-1* and human *Sarm1* can function cell-autonomously to inhibit axon regeneration indicates intrinsic regulation of axon regeneration may be a conserved function of SARM proteins.

*C. elegans* has not been a robust model of motor axon degeneration because in the standard injury model, in which motor axons are cut once, the distal segment remains largely intact in adult animals (*Nichols et al., 2016*). Why the distal segment of severed *C. elegans* motor axons remain intact has been a long-standing point of curiosity (*Figure 7—figure supplement 2*). Here, we found that severing an axon twice creates a middle fragment that completely degenerates and a proximal segment that is capable of regenerating. It follows that since double axotomy does induce degeneration, the resulting injury response may be more representative of the injury responses of other invertebrates and mammals compared to single axotomy. Practically, the double injury assay combines the tractability of *C. elegans* with the ability to investigate motor axon regeneration and degeneration on either side of the injury. The resulting finding that *tir-1* functions cell-autonomously to simultaneously inhibit regeneration of the proximal axon fragment while also promoting degeneration of the distal axon fragment raises a number of questions. One of which is whether the enhanced regeneration in

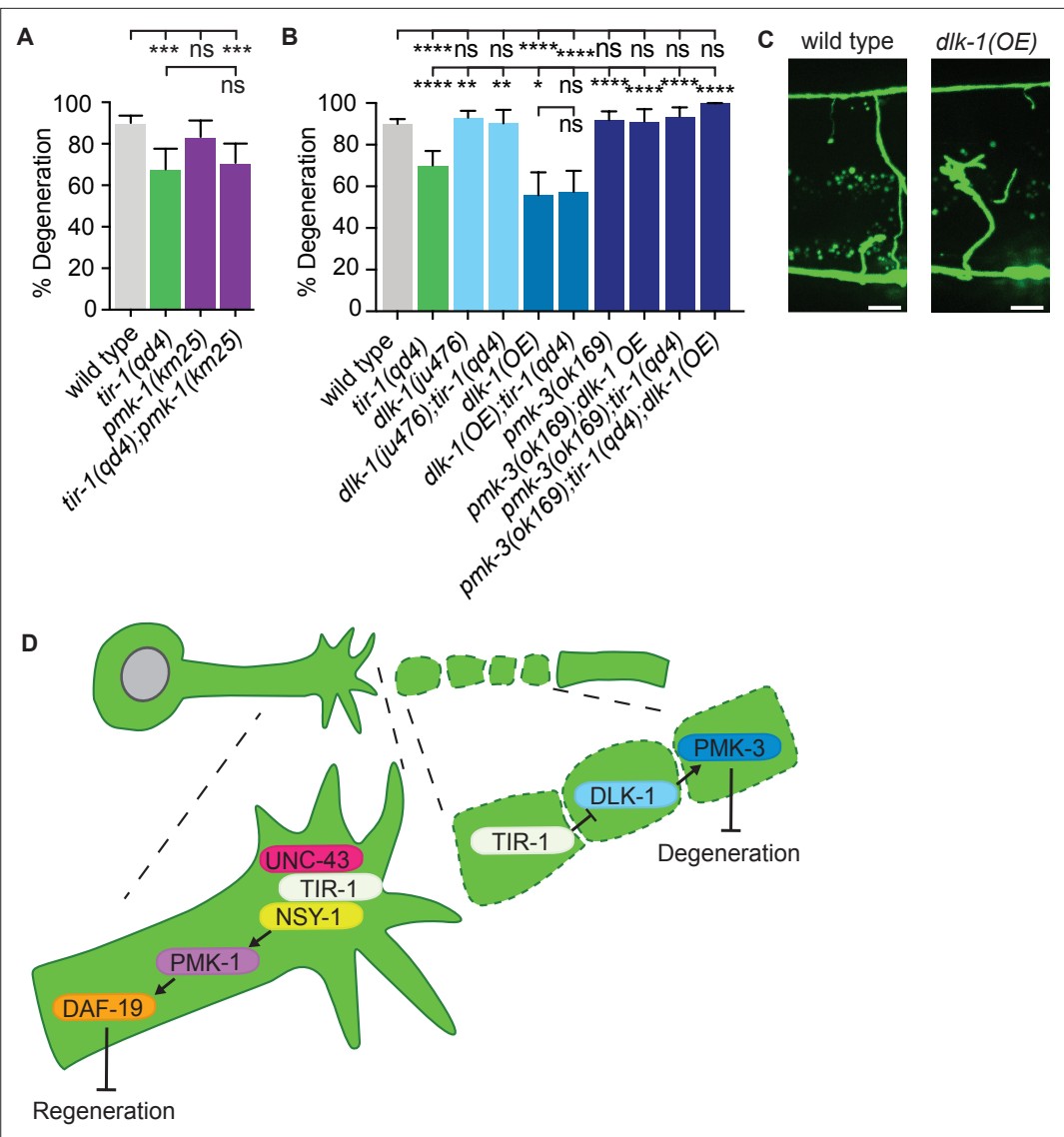

**Figure 7.** TIR-1 functions with the DLK-1 signaling pathway to regulate axon degeneration. (**A**) Loss of *pmk-1/p38* does not affect degeneration with or without *tir-1* function. N = 198, 68, 54, 71. (**B**) Loss of *dlk-1* function suppresses the decreased degeneration of *tir-1(qd4)* animals and overexpression of *dlk-1* in GABA motor neurons inhibits degeneration, indicating DLK-1 negatively regulates axon degeneration. Loss of *pmk-3* function rescues degeneration in all backgrounds, indicating PMK-3 is epistatic to both *dlk-1* and *tir-1*. N = 517, 140, 142, 41, 70, 82, 98, 45, 62, 23. Significance relative to wild type is indicated by *p≤0.05, **p≤0.01, ***p≤0.001, ****p≤0.0001. Error bars represent 95% confidence intervals. (**C**) Representative micrographs of double injured wild type and *dlk-1(OE)* axons in which the respective middle fragments did and did not (asterisk) degenerate. Scale bar: 10 μm. (**D**) TIR-1 inhibits axon regeneration and promotes axon degeneration in the same injured axon. In the proximal fragment, TIR-1 functions with the NSY-1 signaling cascade to modify gene transcription and inhibit axon regeneration. In the severed fragment, TIR-1 promotes degeneration by antagonizing the inhibitory DLK-1 signaling cascade. Significance relative to wild type or *tir-1(qd4)* is indicated by *p≤0.05, **p≤0.01, ***p≤0.001. Error bars represent 95% confidence intervals.

The online version of this article includes the following figure supplement(s) for figure 7:

**Figure supplement 1.** Expression of endogenous *dlk-1* mRNA did not significantly differ between *tir-1(-)* and wild-type animals.

**Figure supplement 2.** DLK-1 does not inhibit distal axon fragment degeneration after a single injury.

*tir-1* mutants is not simply a secondary consequence of reduced degeneration. Either reducing the function of intrinsic inhibitors or changing the amount of extrinsic cues in the environment, such as degeneration and inflammation, can promote axon regeneration (*Curcio and Bradke, 2018*; *He and Jin, 2016*; *Schwab and Strittmatter, 2014*; *Tedeschi and Bradke, 2017*; *Tran et al., 2018*). Here, we found that axons lacking *tir-1* regenerate more often than wild-type axons after a single injury even though a single injury does not cause degeneration in either wild-type or *tir-1* mutants. Thus, the increased regeneration of *tir-1* mutants is not a secondary consequence of a decrease in degeneration. Combined with our finding that *tir-1* functions within GABA axons to regulate axon regeneration, our data suggest that *tir-1* functions within the proximal fragments of injured axons to inhibit axon regeneration independently from its role in degeneration.

## TIR-1/SARM1 inhibits axon regeneration with the NSY-1/ASK1 MAPK signaling pathway

Our finding that *tir-1* regulates both axon regeneration and degeneration raises the question of how this single gene differentially regulates opposite functions in what was once the same axon. Our data indicate that TIR-1/SARM1 carries out these opposing functions by interacting with two divergent MAP kinase signaling pathways on either side of the injury. On the proximal side, TIR-1/SARM1 functions with UNC-43/CAMKII, the NSY-1/ASK1–PMK-1/p38 MAPK signaling cascade, and the transcription factor DAF-19/RFX to inhibit axon regeneration. The involvement of DAF-19 suggests that TIR-1 activation may change the transcriptional profile of the proximal fragment into one that does not support repair. Unlike the TIR-1–NSY-1–PMK-1 pathway that regulates the innate immune response and serotonin biosynthesis, DAF-19 does not require the co-transcription factor ATF-7 to regulate axon regeneration (*Xie et al., 2013*). Therefore, TIR-1 likely achieves specificity of function by regulating the activity or expression of particular transcription factors in a temporal and tissue-specific manner in response to injury. SARM1 has been found to regulate a c-Jun-mediated transcriptional program and an immune response in injured mammalian axons (*Wang et al., 2018*). Whether it could also regulate cytoskeletal dynamics and regeneration in specific cellular contexts through RFX-mediated transcription is an interesting question.

The additional finding that axon regeneration is not affected by mutation in the predicted NADase residue E788A/E642A suggests that the role of *tir-1* in axon regeneration does not depend on its ability hydrolyze $NAD^+$. TIR-1's ability to regulate regeneration in coordination with NSY-1 signaling and independently of its NADase activity contrasts with recent findings that dSarm regulates neuromuscular junction formation and glial phagocytosis using both its NADase activity and interaction with Ask1 signaling (*Brace et al., 2022*; *Herrmann et al., 2022*). The NADase-independent function of TIR-1 also contrasts with the findings that dSarm, mSarm, and hSARM1 rely on their ability to hydrolyze $NAD^+$ to regulate axon degeneration (*Brace et al., 2022*; *Herrmann et al., 2022*; *Hsu et al., 2021*). However, the TIR-1 mechanism parallels the mechanism by which dSarm regulates axon transport within uninjured bystander neurons, which occurs independently of its NADase activity and by interacting with the voltage-gated calcium channel Cacophony (Cac) and Ask1/NSY-1 MAPK signaling (*Hsu et al., 2021*). These data indicate that *dSarm* and *tir-1* can regulate multiple components of the injury response independently of $NAD^+$ hydrolysis. Whether UNC-43–TIR-1–NSY-1–PMK-1–DAF-19 signaling inhibits regeneration in *C. elegans* by suppressing axon transport and how the NSY-1/ASK1 pathway differentially regulates its downstream functions are interesting questions of future investigation.

Our finding that TIR-1 regulates regeneration also raises the question of whether TIR-1's role in regeneration is dependent on NMNAT activity. NMNAT antagonizes SARM1 by maintaining $NAD^+$ in the allosteric pocket of SARM1, which inhibits its $NAD^+$ hydrolase activity (*Bratkowski et al., 2020*; *Figley et al., 2021*; *Horsefield et al., 2019*; *Jiang et al., 2020*; *Sasaki et al., 2016*; *Shi et al., 2022*; *Sporny et al., 2020*). In NMAT mutants, $NAD^+$ is replaced by NMN in the allosteric pocket and the altered conformation of SARM1 activates its NADase activity (*Figley et al., 2021*; *Sasaki et al., 2016*; *Shi et al., 2022*). Previous work demonstrates that NMNAT2 inhibits axon regeneration in *C. elegans* mechanosensory axons (*Kim et al., 2018a*). This could lead to the prediction that NMAT-2 might also inhibit regeneration by inhibiting the $NAD^+$ hydrolase activity of TIR-1. However, here we find that TIR-1 inhibits regeneration of motor axons independently of its function as an $NAD^+$ hydrolase, indicating that NMNAT-2 must not inhibit regeneration by suppressing the NADase activity of TIR-1.

Nonetheless, it remains possible that an NMNAT-dependent change in conformation determines whether TIR-1 can interact with downstream effectors of regeneration such as NSY-1. In this model, NMAT-2 would indeed facilitate TIR-1's ability to inhibit regeneration. Alternatively, since many genes differentially regulate axon regeneration in a cell-type-dependent manner (*Kim et al., 2018a*), it is also possible that NMAT-2 and TIR-1 differentially regulate regeneration in the mechanosensory and motor axons.

## TIR-1/SARM1 regulates injury-induced axon degeneration in coordination with the DLK-1 MAPK signaling pathway

On the distal side of the injury, our data suggest that TIR-1 regulates degeneration by interacting with an alternate MAPK pathway. Loss of *pmk-1* function has no effect on axon degeneration, nor does it affect the degeneration phenotype of *tir-1* loss-of-function mutants, indicating that TIR-1 regulates axon degeneration independently from the NSY-1–PMK-1 MAPK pathway. Instead, TIR-1 promotes axon degeneration through interaction with DLK-1–PMK-3 MAPK signaling. Specifically, our data suggest that in *C. elegans*, TIR-1 promotes injury-induced degeneration by inhibiting a protective function of DLK-1 and PMK-3. The protective function of DLK-1 and PMK-3 in axon degeneration appears to be conserved in specific contexts. For example, Wallenda (Wnd), the DLK-1 homolog in *Drosophila melanogaster* inhibits axon degeneration in motor axons that have undergone a conditioning lesion (*Xiong and Collins, 2012*). However, Wallenda also promotes degeneration in singly severed olfactory receptor axons and mammalian DLK promotes degeneration of severed dorsal root ganglion axons (*Ghosh et al., 2011*; *Miller et al., 2009*). While it is unlikely that the small amount of time between axotomies in the double injury model induces a conditioning lesion, the comparison demonstrates that the nature of MAPK activity can be differentially determined by specific types of injury and by cell type. In relation to SARM signaling, SARM1 and dSarm also regulate axon degeneration by interacting with the signaling pathways of DLK-1 homologs Wallenda and DLK (*Summers et al., 2020*; *Walker et al., 2017*; *Yang et al., 2015*). Separate studies have found that MAPK signaling functions downstream of SARM1 to regulate degeneration of crushed retinal ganglion cells and upstream of SARM1 to regulate degeneration of cultured mammalian dorsal root ganglia neurons and *Drosophila* motoneurons (*Summers et al., 2018*; *Walker et al., 2017*; *Yang et al., 2015*). Together with this multi-faceted relationship between SARM1 and MAPK signaling, our data indicate the highly conserved TIR-1, dSarm, and SARM1 proteins regulate injury-induced degeneration through interactions with shared MAPK signaling components. However, the precise interactions between TIR-1/SARM1/dSarm and DLK-1/MAPK/Wnd signaling are complex and likely depend on cell type, form of injury, and species. Identifying the similarities and differences between the various interactions between SARM and MAPK signaling across species and types of injury will be important to understanding how these complex genes regulate the injury response.

In addition to regulating the injury response, TIR-1 is required for degeneration in a *C. elegans* model of ALS, functions in the epidermis to promote age-dependent degeneration of PVD dendrites, induces spontaneous degeneration when forced to multimerize, and inhibits spontaneous axon degeneration of PVQ interneurons in mutant *ric-7(-)* animals, which lack mitochondria (*Ding et al., 2022*; *Lezi et al., 2018*; *Loring et al., 2021*; *Rawson et al., 2014*; *Vérièpe et al., 2015*). Questions raised specifically from a comparison of the opposite destructive and protective roles of TIR-1 in response to injury and absence of mitochondria include whether a specific injury signal determines that TIR-1 takes on a destructive role and whether TIR-1 can protect multiple types of neurons against the spontaneous degeneration that is induced by the absence of mitochondria. Looking downstream, we found the map kinase *pmk-3* is epistatic to *tir-1* and inhibits injury-induced axon degeneration. Interestingly, *pmk-3* was also found to inhibit spontaneous degeneration in the absence of mitochondria, suggesting the difference between promotion and suppression of degeneration may ultimately be attributed to differential regulation of PMK-3 in these two contexts.

To date, how SARM1 ultimately leads to axon degeneration after injury is not completely clear. Besides MAP kinase signaling, effectors of dSarm and SARM1 function include diminished levels of NAD$^+$ and the downstream Axundead protein (*Essuman et al., 2017*; *Neukomm et al., 2017*). *C. elegans* does not have an identifiable Axundead homolog; however, the NADase activity of SARM1 and dSarm is conserved, albeit perhaps to a weaker extent, in the *C. elegans* TIR domain (*Horsefield et al., 2019*; *Loring et al., 2020*; *Neukomm et al., 2017*; *Peterson et al., 2022*; *Summers et al.,*

*2016*). Our finding that TIR-1 can promote injury-induced degeneration despite having a E788A/E642A mutation in its key catalytic residue, along with a recent finding that E788A renders the *C. elegans* TIR domain catalytically dead and is required for TIR-1-mediated pathogen avoidance, indicates that although TIR-1 does function as an NADase, it regulates injury-induced axon degeneration independently of its NADase activity. How the D773A/D627A mutation disrupts TIR function remains an open question (*Loring et al., 2020*; *Summers et al., 2016*). It is possible that in addition to disrupting the enzymatic activity of TIR-1, D773A also disrupts one of the many complex intramolecular and intermolecular interactions of TIR-1, which suppresses degeneration by disrupting a yet unidentified mechanism of action. Knowing whether TIR-1 induces injury induced degeneration through direct interaction with DLK-1 will contribute significantly to our understanding of how TIR-1 promotes NADase-independent degeneration.

While TIR-1 can promote NADase-independent degeneration, dSarm and SARM1 depend on NAD$^+$ hydrolysis to regulate injury-induced degeneration (*Brace et al., 2022*; *Herrmann et al., 2022*; *Hsu et al., 2021*). Interestingly, while overexpressing wild-type *tir-1b* in motor neurons induced spontaneous degeneration, expressing the same amount of mutant *tir-1b*, containing the D773A or E788A mutation, did not induce significant degeneration. Therefore, while TIR-1 does not require its NADase activity to promote injury-induced degeneration, it can use its enzymatic activity to induce spontaneous axon degeneration. These data suggest that TIR-1 has evolved dual mechanisms to regulate degeneration in response to different insults that it encounters in the wild.

The finding that TIR-1, a homolog of dSarm and SARM1, essential regulators of axon degeneration, also regulates the opposite response to injury, axon regeneration, is unexpected. In addition, TIR-1 was not thought to regulate injury-induced degeneration of *C. elegans* axons, likely because adult motor axons do not robustly degenerate using the single-injury assay and because degeneration of injured mechanosensory axons is regulated by apoptotic engulfment machinery in surrounding tissues, and not by TIR-1 (*Byrne et al., 2011*; *Nichols et al., 2016*). The identified dual role of TIR-1 in the injury response raises a number of compelling questions, including: Is TIR-1 differentially activated on either side of the injury to determine whether a fragment repairs itself or self-destructs? Does TIR-1 regulate regeneration throughout the nervous system, or, like other intrinsic regulators of regeneration, does it function according to cell type and age (*Byrne et al., 2014*; *Duan et al., 2015*; *Geoffroy et al., 2016*)? Recent studies support the idea that SARM orthologs might function intrinsically to directly regulate axon regeneration. Systemic inhibition of *tir-1/Sarm1* increases axon regeneration after injury in zebrafish. In addition, a mammalian study suggests functional recovery and axon regeneration after injury is improved in the absence of SARM1 function in neurons and in astrocytes, in part due to reduced inflammation and increased protection of injured axons (*Asgharsharghi et al., 2021*; *Liu et al., 2021*). This data is supported by a preceding finding that SARM1 function within injured axons regulates an immune response that triggers inflammation (*Wang et al., 2018*). Another recent study demonstrated that SARM1 knockout mice have more collateral axon branching and increased cytoskeletal dynamics in postnatal sensory neurons, indicating that SARM1 inhibits axon growth even in the absence of injury (*Ketschek et al., 2022*). In light of these results, our finding that GABA-specific TIR-1 and hSARM1 expression inhibits axon regeneration even in animals that lack extrinsic inflammatory cells raises the intriguing question of whether SARM1 signaling can be manipulated to promote axon regeneration by directly driving cytoskeletal growth in injured mammalian axons.

In conclusion, we present a model where TIR-1 functions cell-autonomously in GABA motor neurons to regulate axon regeneration. We show that to achieve specificity TIR-1 directs the opposite processes of axon regeneration and axon degeneration by interacting with divergent downstream MAPK signaling pathways on either side of the injury. Understanding how TIR-1 regulates a balance between axon regeneration and degeneration may provide novel ways to push the nervous system toward regeneration in response to injury.

# Materials and methods

## *C. elegans* strains and culture

Unless otherwise noted, all *C. elegans* strains were maintained at 20°C on NGM plates seeded with OP50 *Escherichia coli*. Strains listed below were kindly provided by the Alkema, Francis, Hammarlund, Pukkila-Worley, Hanna-Rose labs, and the *Caenorhabditis* Genetics Center, which is funded by the NIH

Office of Research Infrastructure Programs (P40 OD010440). To visualize GABA motor neurons, strains were crossed into EG1285 (*oxIs12 [unc-47p::GFP, lin-15(+)]*). Mutants used in this study: *tir-1(qd4)*, *tir-1(tm3036)*, *ufIs175(flp-13p::myristoylatedGFP)*, *nsy-1(ag3)*, *pmk-1(km25); unc-43(e408)*, *daf-19(m86)*, *dlk-1(ju476)*, *pmk-3(ok169)*, *pnc-1(ku212)*, *tol-1(nr2033)*, *ced-1(e1735)*, *atf-7(gk715)*.

## Laser axotomy, imaging, and quantification

Axotomy was performed as previously described (*Byrne et al., 2011*), unless otherwise stated below. Larval stage 4 (L4) worms were mounted on 3% agarose pads and immobilized with microbeads (Polysciences) or 0.1 mM levamisole. Single injuries were made at the midline of the axon commissure and 1–3 axons were cut per worm. Double injuries were made with two targeted cuts, one roughly halfway between the ventral nerve cord and the midline of the worm and the second roughly halfway between the midline of the worm and the dorsal nerve cord, creating an approximately 25 μm middle fragment. The two injuries are performed within seconds of each other in the same order (proximal-distal). One axon was cut per worm in the double injury experiments. Axon regeneration was quantified 24 hr after injury unless otherwise stated by placing animals immobilized with microbeads or 0.1 mM levamisole on a 3% agarose pad. Animals were imaged with a Nikon 100x 1.4 NA objective, Andor Zyla sCMOS camera and Leica EL6000 light source. Regeneration was scored as the percent of axons that regenerated fully to the dorsal cord (F), 75% of the distance to the dorsal cord (M$^+$), to the midline (M), to below the midline or had 3+ filopodia extending from the proximal segment (M$^-$). Axons that failed to regenerate or had two or fewer small filopodia protruding from the proximal segment were scored as N and NF, respectively. To control for day-to-day and instrument-to-instrument variability, experiments were carried out with same day controls on one axotomy rig.

## CRISPR

CRISPR gene editing was conducted as previously described (*Dokshin et al., 2018*). crRNA, tracrRNA, and Cas9 were obtained from IDT and injected with the *rol-6* co-injection marker. CRISPR mutants were genotyped and outcrossed before analysis.

## TIR-1 expression

We generated transgenic lines expressing TIR-1 under its endogenous promoter by expressing bamEx102[*tir-1p::tir-1b::mCherry::unc-54 3'UTR+ceh-22::gfp]* in ABC16 (*tir-1(qd4); oxIs12[unc-47p::GFP, lin-15(+)]*) animals using standard microinjection (*Mello et al., 1991*). L4 stage animals were imaged on Perkin Elmer Precisely UltraVIEW VoX confocal imaging system. To determine the subcellular localization of TIR-1, we generated transgenic animals expressing *unc-47p::mScarlet::tir-1b:: let-858 3'UTR* and co-injection marker *ceh-22::gfp* in EG1285 (*oxIs12[unc-47p::GFP, lin-15(+)]*) animals. To determine the tissue specificity of TIR-1 function, we generated transgenic animals expressing *unc-47p::mScarlet::tir-1b::let-858* 3'UTR and co-injection marker *pstr-1::gfp* in ABC16 (*tir-1(qd4); oxIs12 [unc-47p::GFP]*) animals. To determine whether TIR-1 promotes chronic axon degeneration, we expressed *unc-47p::tir-1b::mCherry::unc-54 3'UTR* and co-injection marker *pstr-1::gfp* in EG1285 (*oxIs12[unc-47p::GFP, lin-15(+)]*) animals. Transgenic lines were established using standard protocols (*Mello et al., 1991*). L4 stage animals were imaged on Perkin Elmer Precisely UltraVIEW VoX confocal imaging system.

## hSARM1 expression

To determine whether axon regeneration is a conserved function of SARM1, we built transgenic animals expressing mScarlet::SARM1 from the *unc-47* GABA-specific promoter. The SARM1 plasmid was a kind gift from the Fitzgerald lab at UMass Medical School. The SAM-TIR domain of SARM1 was cloned into the *unc-47p::mScarlet::tir-1b::let858 3'UTR* plasmid. The resulting *unc-47p::mScarlet::SARM1::let858 3'UTR* plasmid was injected into ABC16 (*tir-1(qd4) III; oxIs12 X*) and *oxIs12(unc-47p::GFP X)* animals. The plasmid was injected at a final concentration of 10 ng/ul to generate bamEx4070, bamEx4072, bamEx4073. Further we amplified a 1.286 kb band spanning the end of the mScarlet sequence into the beginning of the *let-858* 3' UTR using PCR to confirm expression within these animals (primers: GCCGACTTCAAGACCACCTACAAG and CAACGACGTTGGCGTCGATC).

## Injury-induced degeneration

Middle fragment degeneration after double injury was quantified 24 hr after injury unless otherwise indicated and grouped into three categories: degenerated (complete clearance), partially degenerated

(>65% of the middle fragment degenerated), or largely remained intact (>80% of middle fragment remained) 24 hr after double injury. Chronic degeneration in the absence of injury was assessed by immobilizing L4 stage worms with 0.1 mM levamisole on a 3% agarose pad. Images were taken on a Perkin Elmer Precisely UltraVIEW VoX spinning disc confocal imaging system using a 40x objective. Images were compressed and exported as TIFF files for processing in FIJI. The axon or dorsal nerve cord segment to be analyzed was traced using a segmented line ROI of fixed size and mean intensity was quantified.

## NADase point mutations

To determine whether the NADase function of TIR-1 is required to induce chronic degeneration, we built transgenic animals expressing either *unc-47p::tir-1b(E788A)::mCherry::unc-54 3'UTR* or *unc-47p::tir-1b(D773A)::mCherry::unc-54 3'UTR* plasmids. Point mutations were generated using Quik-Change site directed mutagenesis on the *unc-47p::tir-1b::mCherry::unc-54 3'UTR* plasmid. Primers used to create the E788A point mutation were GCACAAAGCGCTGAAATGTGCATTTG and CATT TCAGCGCTTTGTGCACCCAATCC. Primers used to generate the D773A point mutation were CAGTTTGGCTAGACTTTTAAATGATGATAACTG and GTCTAGCCAAACTGTTTGGTGTGAG. The resulting plasmids were injected into ABC16 (*tir-1(qd4) III; oxIs12 X*) animals at 50 ng/ul with *ceh-22::GFP* as a co-injection marker to generate bamEx4003[*unc-47p::tir-1b(E788A)::mCherry::unc-54 3'UTR+ceh-22::GFP*], bamEx4052[*unc-47p::tir-1b(E788A)::mCherry::unc-54 3'UTR+ceh-22::GFP*], bamEx4049[*unc-47p::tir-1b(D773A)::mCherry::unc-54 3'UTR+ceh-22::GFP*], bamEx4050[*unc-47p::tir-1b(D773A)::mCherry::unc-54 3'UTR+ceh-22::GFP*] animals. The number of intact, damaged, and missing GABAergic axons was quantified by mounting L4 stage animals on 3% agarose pads in 300 nM sodium azide. Axons were scored as damaged if they were broken or if there was significant beading. All other axons were scored as intact if they traversed completely from the ventral nerve cord to the dorsal nerve cord. The number of cell bodies was counted starting at the tail moving anteriorly toward the head. We observed an average of 15 axons as the 16th most anterior axon was either missing or redirected to the left side of the animal.

## TIR-1b fluorescence intensity quantification

To determine TIR-1b levels in animals expressing *unc-47p::tir-1b::mCherry::unc-54 3' UTR*, *unc-47p::tir-1b(E788A)::mCherry::unc-54 3' UTR*, and *unc-47p::tir-1b(D773A)::mCherry::unc-54 3' UTR*, we quantified fluorescent expression in the posterior seven GABAergic cell bodies. These cell bodies correspond with axons that were severed in the injury assays. Images were taken on a Perkin Elmer Precisely UltraVIEW VoX spinning disc confocal imaging system using a 40x objective. Images were compressed and exported as TIFF files for processing in FIJI. Cell bodies were then traced using the freehand tool in FIJI and the intensity of TIR-1b::mCherry was quantified. Background fluorescence was corrected by quantifying an additional trace next to each cell body.

## RT-qPCR

Strains analyzed: *oxIs12(Punc-47::GFP), tir-1(qd4);oxIs12, dlk-1(oe);oxIs12, dlk-1(oe);tir-1(qd4);oxIs12*. RNA isolation: 100 L4 stage animals of each strain were transferred to unseeded NGM plates and transferred to 50 µl RNase-free dH$_2$O. 500 µl of cold TRIZOL was added and samples were frozen in a –80°C freezer overnight. The following day, another 500 µl of cold TRIZOL reagent was added to each sample. Samples were subjected to three rounds of freeze-thaws that included flash-freezing in liquid nitrogen, thawing in 37°C water bath, and vortexing for 30 s. Samples were passed through 18-Gauge syringes 12 times and centrifuged at 14,000 rpm for 10 min at 4°C to remove insoluble material. Supernatant was transferred to fresh microcentrifuge tubes and 200 µl of CHCl$_3$ was added. Tubes were inverted and vortexed for 30 s, incubated at room temperature for 5 min, and centrifuged at 14,000 rpm for 15 min at 4°C to separate phases. 500 µl of the clear, upper aqueous phase was transferred to fresh microcentrifuge tubes and 500 µl isopropanol was added and mixed by pipetting. RNA was isolated with QIAGEN RNeasy purification and eluted with 36 µl of RNase-free dH$_2$O. cDNA synthesis: reverse transcriptase (RT) synthesized cDNA, and No-RT samples were prepared with NEB LunaScript RT SuperMix kit. The concentration of each was then quantified using a NanoDrop. RT-qPCR analysis: Procedures were followed according to the NEB Luna Universal qPCR kit protocol.

Reactions were performed in triplicate, with RT+ template and No-RT template control. RT-qPCR was performed with a Bio-Rad CFX96 Real-Time System with a C1000 Touch Thermal Cycler.

## Statistical analysis

Statistical analysis was performed using GraphPad QuickCalcs (https://www.graphpad.com/quick-cals/) and Prism (GraphPad). Categorical data: Data from repeated assays comparing a mutant and corresponding control strain were pooled for statistical analysis. Bars represent 95% confidence intervals and significance determined with Fisher's exact test, where $*p \leq 0.05$, $**p \leq 0.01$, $*** \leq 0.001$, $n \geq 30$. Continuous data (fluorescence intensity of GFP along axon commissure) are presented as the mean with error bars representing the standard error of the mean. Significance was calculated with Student's $t$-tests, where $*p \leq 0.05$, $** \leq 0.01$, $*** \leq 0.001$.

## Materials availability

All strains and reagents used in the analysis are available upon request.

## Acknowledgements

We thank A Zeamer for technical contribution, W Joyce and M Liu for construction of reagents, Craig Mello and Krishna Ghanta for help with CRISPR design and injections, K Fitzgerald for the hSARM1 plasmid, P Thompson, H Loring, J Icso, R Pukkila-Worley, N Peterson, as well as the M Francis and M Alkema labs for helpful discussions, along with M Alkema, P Emery, and V Budnik for helpful comments on the manuscript.

## Additional information

### Funding

| Funder | Grant reference number | Author |
|---|---|---|
| National Institute of Neurological Disorders and Stroke | 5R01NS110936 | Alexandra B Byrne |
| National Institute of Neurological Disorders and Stroke | 1F31NS124338 | Lauren C O'Connor |
| The Dan and Diane Riccio Fund for Neuroscience | | Alexandra B Byrne |

The funders had no role in study design, data collection and interpretation, or the decision to submit the work for publication.

### Author contributions

Victoria L Czech, Conceptualization, Formal analysis, Validation, Investigation, Visualization, Writing – original draft, Writing – review and editing; Lauren C O'Connor, Conceptualization, Formal analysis, Validation, Investigation, Visualization, Writing – review and editing; Brendan Philippon, Formal analysis, Investigation; Emily Norman, Formal analysis, Investigation, Writing – review and editing; Alexandra B Byrne, Conceptualization, Formal analysis, Supervision, Funding acquisition, Investigation, Visualization, Methodology, Writing – original draft, Project administration, Writing – review and editing

### Author ORCIDs

Lauren C O'Connor http://orcid.org/0000-0002-1966-6932
Emily Norman http://orcid.org/0000-0002-1575-3473
Alexandra B Byrne http://orcid.org/0000-0002-7449-9188

### Decision letter and Author response

Decision letter https://doi.org/10.7554/eLife.80856.sa1
Author response https://doi.org/10.7554/eLife.80856.sa2

# Additional files

## Supplementary files
• MDAR checklist

## Data availability

All numerical data analysed during this study are included in the main text and figure legends. No additional datasets were generated.

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
