## [Editor Report]

This important study reveals a new role for TIR-1/SARM1 in inhibiting both axonal regeneration and promoting axon degeneration in a novel axonal injury model. The authors provide solid evidence to support their conclusions that TIR-1 mediates these processes by acting via distinct signaling pathways. This work will be of general interest to developmental neuroscientists.

---

## [Decision Letter]

**Decision letter after peer review:**

[Editors’ note: the authors submitted for reconsideration following the decision after peer review. What follows is the decision letter after the first round of review.]

*Reviewer #1:*

Julian and Byrne use the *C. elegans* interneurons as a model to test the involvement of TIR-1 in axon regeneration. TIR-1's function in axon degeneration is well established but its function in regeneration is novel. The authors show convincingly that TIR-1 inhibits axon regeneration through the NSY-1 MAPK pathway. Since TIR-1 is a NAD+ degradation enzyme, the authors should test if overexpression of NMnat can also increase regeneration. The following issue should also be addressed.

For figure 3, what causes the GFP levels to reduce in the pGABA::tir-1? mCherry protein fusion is often problematic because it forces proteins to aggregate. In Figure 2C, it seems that significant amount of GFP colocalized with the mCherry puncta. The authors should use GFP to fuse with tir-1 and use a cytosolic mCherry as a marker to repeat this experiment. At present, the evidence that TIR-1 overexpression causes degeneration is not strong. mCherry protein fusions sometimes cause aggregates and toxicity. Experiments in Figure 5D should be repeated with tir-1GFP or untagged tir-1.

*Reviewer #2:*

Byrne and colleagues examine whether the major executioner of Wallerian degeneration, Sarm-1, serves this function in *C. elegans* and has an additional role in axon regeneration. They make use of a well-established system to study axotomy-induced axon regeneration and present a novel paradigm to cause injury-induced degeneration in these same neurons. The authors present evidence that Tir-1, the presumed Sarm1 ortholog in *C. elegans*, has a novel role to inhibit axon regeneration upon injury and suggest that Tir-1 promotes degeneration and inhibits regeneration via distinct MAPK pathways. Inhibition of Sarm1 has garnered significant attention to block neurodegeneration in several contexts. Therefore, the implications of this result – that Sarm1 inhibition may have the previously unknown effect of enhancing an adaptive regenerative response – are large and would be of significant interest in the field. Yet much of the excitement about this work (and certainly the tone of the writing) assumes that Tir-1 functions in *C. elegans* as it does in flies and mammals, as the key executioner of Wallerian degeneration which is not resolved by the experiments presented.

I have several big-picture concerns about this work:

1) Does TIR-1 promote axon degeneration similarly to dSarm/Sarm1?:

– Are the authors studying bona fide Wallerian degeneration? Loss of dSarm/Sarm1 potently and completely protects severed axons; 100% protection, lasting for days. The authors use a novel two-cut paradigm to allow for injury-induced degeneration (since injured axons do not normally degenerate in *C. elegans*), but only observe a ~20% protective effect (Figure 3A) in the qd4 allele, and only at a single time point. This is very different from what would be expected of a true dSarm/Sarm1 ortholog, and a key point in interpreting this work in the context of the broader Sarm1 literature. Is this two-cut degeneration dependent on NAD levels as would be expected for degeneration regulated by dSarm/Sarm1? Or is this degeneration mechanistically distinct? This could easily be tested by overexpression of NMNAT enzymes or mutation of the conserved NAD-hydrolyzing domain of Tir-1, both of which should protect severed axons if the authors' model is correct.

2) Does the genetic pathway for Tir-1-dependent regeneration fit their data?

– Loss of Dlk-1 and Pmk-3 both suppress the enhanced regeneration seen in Tir-1 mutants (Figure 4C). Further, there appears to be no genetic interaction between Tir-1 and Nsy-1/Pmk-1 (Figure 4B), and the authors state that the nsy-1-pmk-1 MAPK pathway "does not act synthetically or in parallel" to the dlk-1-pmk-3 MAPK pathway in tir-1 function in axon regeneration. Despite these data, the authors' model (Figure 5E) is that Tir-1 inhibits regeneration through Nsy-1/Pmk-1, not via Dlk-1/Pmk-3. I do not clearly see how these epistasis experiments support their model.

*Reviewer #3:*

Julian and Byrne investigate the function of TIR1, the worm ortholog of SARM1, in the axon injury response. In worms TIR1 participates in a number of signal transduction pathways, while in mammals and flies SARM1 is best known for its essential role in Wallerian degeneration. Here, Julian and Byrne find that TIR1 inhibits axon regeneration using its canonical nsy-1/Ask1 signaling pathway. Worms do not undergo Wallerian degeneration, however the authors develop a double axon injury model where an internal axon segment is disconnected on both its proximal and distal sides. These axon fragments do degenerate, and TIR1 mildly promotes this degeneration. This TIR1-dependent component of degeneration requires a different MAP kinase cascade than does regeneration. The authors conclude that in worms TIR1 regulates both axon regeneration and axon degeneration via distinct mechanisms.

This is a solid manuscript that clearly demonstrates that TIR1 is a negative regulator of axon regeneration in worms. Many genes have been identified to regulate axon regeneration in this system, and so this is an interesting advance that will be useful to workers in the field attempting to build a comprehensive understanding of *C. elegans* axon regeneration. The significance of the findings with degeneration of doubly injured axon fragments is less clear. TIR1 does not trigger degeneration of distal axons in the worm as it does in mice and fish and flies, and even the effect seen here with the double-injured axon fragments is quite modest (86% of WT fragments degenerate while 68% of TIR1 mutant fragments degenerate). In contrast, SARM1 is absolutely required for degeneration in these other organisms. Moreover, the DLK signaling pathway identified is well known to participate in axon degeneration in other organisms. As such, the advance for the field of axon degeneration is modest. Indeed, it would be much more significant to understand why TIR1 is unable to trigger axon degeneration in worms rather than to show this modest effect in this double injury model.

There are a few additional issues for the authors to consider.

1) The abstract and introduction of the story is framed as a choice between axon regeneration and degeneration that must be balanced. However, there is no conflict between the two because they occur in different axonal compartments (proximal vs distal). Indeed, in the mammalian PNS, axon degeneration and regeneration work together for an effective injury response. Axon degeneration is necessary to clear the distal axon and allow for axon regeneration.

3) The authors mention that overexpression of SARM1 promotes axon degeneration. This is an interesting finding. The authors should assess whether overexpression phenotype induces cell death or is truly selective axon degeneration.

[Editors’ note: further revisions were suggested prior to acceptance, as described below.]

Thank you for resubmitting your work entitled "TIR-1/SARM1 Inhibits Axon Regeneration" for further consideration by *eLife*. Your revised article has been evaluated by Piali Sengupta (Senior/Reviewing Editor) as well as two of the previous reviewers.

The manuscript is much improved but there are some remaining issues that need to be addressed, as outlined below. We realize that you have already done a lot of experiments at this point, so it is up to you whether you would like to perform additional experiments as suggested below.

Essential revisions:

1 While a subset of experiments addressing degeneration are performed after the single axotomy, others are performed after a double cut. It is unclear whether the mechanisms underlying degeneration in these different experimental paradigms is identical or different. Specifically, the authors should compare the role of the MAPK components for degeneration between the single and double-cut protocols.

2. The role of DLK-1/TIR-1 in axon degeneration needs to be clarified with further experiments. The data do not always support the conclusions and in some cases, the two contradict each other – please see points 8-10 from Reviewer 2.

3. An NAD-independent function for TIR-1 in promoting degeneration is of great interest but this requires further substantiation, particularly in the context of being used in the somewhat unusual double-cut injury model. Does the expression of NADase-dead activated TIR-1 in the single cut model and overexpression of TIR-1 in the spontaneous degeneration model promote degeneration?

*Reviewer #1 (Recommendations for the authors):*

Czech and colleagues present work showing that TIR-1 inhibits axon regeneration in *C. elegans* while also promoting degeneration in response to a double axon injury. They have adequately addressed the questions raised in the first round of review, and I feel that their work will attract broad interest in the field. Yet as they have added new and fascinating data, new questions necessarily arise. Below are several points that I feel the authors should clarify:

1) In Figure 2E, F the columns in the graph seem to be mislabeled and prevent accurate assessment of the data. Both present rescue experiments comparing injury-induced regeneration in WT, tir-1(qd4), and then what appear to be rescue experiments (qd4 + tir-1 in 2E, qd4 + Sarm1 in 2F). The third and fourth bars in each are labelled the same. Inferring from the text, perhaps one is a full-length cDNA and the other is a constitutively active form? The Figure legend does not clarify this.

2) The authors show a fascinating result – that tir-1-dependent axon degeneration does not require the NAD hydrolayse activity, thus making tir-1's pro-degenerative role quite unique. However, they cite a Neuron paper from Marc Freeman as a precedent for NAD-independent pro-degenerative roles of dSarm. This citation is misleading. In the Freeman paper, they show that axons adjacent to those degenerating in a classical NAD-dependent manner have cell biological defects (trafficking, etc.) that are dSarm-dependent but NAD-independent. In other words, Freeman does not show that dSarm kills axons in a NAD-independent manner. The authors should clarify this citation, but also, importantly, clarify in their text that the pro-degenerative role of tir-1 has no current parallel in other systems.

3) Regeneration is initially assayed after a single injury, for example in Figure 1 where the original tir-1 qd4 phenotype is shown. But by the time pathway analysis is performed in Figure 5 (showing genetic interactions with the MAPK pathway), regeneration is now characterized after double injury. The authors make no attempt to reconcile this very strange switch in paradigm. Was it simply out of convenience that these were the same animals assayed for injury-induced degeneration after double-cut? Regardless, the authors should show that the regenerative response (timing, magnitude, genetic dependences) are the same (or different?) between single and double cuts.

*Reviewer #2 (Recommendations for the authors):*

I commend the authors on making extensive revisions. The demonstration of a role for Tir-1 functioning through the Ask1 pathway to regulate axon regeneration is clear, although would be solidified with the genetic rescue. Unfortunately, despite the new data, the findings on Tir-1 and axon degeneration remain difficult to follow. The double cut system is of unclear physiological relevance, key experiments yield mechanistically distinct results that are not clearly addressed by the authors, and whatever Tir-1 does do for axon degeneration in *C. elegans* is quite different than its role in other characterized organisms and so the broader significance is questionable.

2) In Figure 2E what is the impact on the regeneration of overexpressing tir-1 in a wild-type background? Overexpressing tir-1 is only shown in a Tir-1 mutant background. However, overexpression of tir-1 may cause inhibition of regeneration even in wild type, which would then mean the data in the mutant background cannot be cleanly interpreted as a rescue.

4) 2F: Is a deletion of the autoinhibitory domain of human Sarm1 being expressed? Expression of such a construct in flies or mammalian cells leads to cell death. Could this just be making the cell sick so it can't regenerate as well? What is the influence of expressing activated human Sarm on regeneration in WT?

5) Figure 3: The key demonstration that Tir-1 promotes axon degeneration in worms is the finding that double-cut middle axon fragments degenerate slightly less in tir-1 mutant than in wild type. This key result is not rescued and should be.

7) Figure 4: I am assuming that in this figure an activated Tir deletion mutant is being used. If so, this is well known to cause toxicity in other systems due to unrestrained enzymatic activity. The authors find that overexpression of this toxic protein promotes axon degeneration when the axons are cut. The authors make the claim that the degenerative mechanism in worms is not NADase-dependent. The assay here is somewhat similar to what has been done in other systems, where the effect of Sarm1 is always enzyme-dependent. Does this prodegenerative effect require the NADase activity? It is unclear whether the different axon degeneration models used in this paper are testing the same mechanism of Tir-1. Assessing the NADase activity here would help establish whether or not Tir-1 is having a consistent effect in these different assays.

8) Figure 6: The role of DLK/Tir-1 is unclear and does not agree with other data in the paper. The authors show that DLK OE leads to less AxD (Figure 6D), and this decrease in AxD requires Tir-1 (suppressed by loss of Tir-1, so in the absence of Tir-1 there is more degeneration). This means that DLK is inhibiting axon degeneration and this effect requires Tir-1, and so Tir-1 is functioning to decrease AxD. However, this is in direct contradiction to the wild-type case, where a loss of function Tir-1 mutant leads to less axon degeneration, arguing that Tir-1 promotes AxD. This highlights how unclear the findings are with Tir-1 and axon degeneration. In one case Tir-1 promotes and in the other case it inhibits degeneration.

[Final comments from Reviewers.]

Reviewer #1 (Recommendations for the authors): The authors have done a commendable job addressing the serious concerns of the reviewers. While the relevance of the double-cut model to the broader field of Wallerian degeneration remains to be determined, these data will nevertheless spark important discussions in the field.

Reviewer #2 (Recommendations for the authors): I commend the authors on their thorough revisions. I know that we asked for a lot in these multiple rounds of revision. While I am sure this was frustrating, I believe it has made for a much stronger paper. All of my original concerns have been fully addressed.

---

## [Author Response]

[Editors’ note: the authors resubmitted a revised version of the paper for consideration. What follows is the authors’ response to the first round of review.]

Whether TIR-1 regulates axon regeneration by depleting NAD^+^ levels is an important question since TIR-1 and its SARM1 orthologs are well-characterized NAD^+^ hydrolases^1-8^. To directly address whether TIR-1 regulates regeneration by hydrolysing NAD^+^, we CRISPR edited the endogenous *tir-1* gene to create E788A and D773A mutations, which correspond to E642A and D627A mutations in SARM1^1-3,7^. E788/E642 is a key catalytic glutamate that is required for the NADase activity of both the SARM1 and TIR-1 TIR domains^1-4,6,7^. The D773/D627 aspartate also promotes NAD^+^ catalysis and is required for the degeneration activity of SARM1^1,3^. We found that both *tir-1(D773A)* and *tir-1(E788A)* mutant animals displayed wild type levels of axon regeneration, indicating TIR-1^D773A^ and TIR-1^E788A^ are functionally expressed and that the NADase activity of TIR-1 is not required for its ability to inhibit axon regeneration. Together, our data reveal that TIR-1 inhibits axon regeneration by interacting with the UNC-43/CAMKII – NSY-1/ASK1 MAPK – PMK1/p38 MAPK – DAF-19/RFX1-3 pathway, and independently from its NADase activity. We have added this result to the manuscript at Line 229 and in Figure 5A.

Multiple lines of evidence indicate that overexpression of TIR-1 induces axon degeneration. As suggested by the reviewer, we expressed *tir-1::GFP* or untagged *tir-1* in GABA neurons and found that these constructs also induce degeneration (see Author response table 1). Note that each line is derived from a separate injection of *tir-1* and therefore each express unique extrachromosomal arrays and number of *tir-1* coding sequences. In contrast, expression of either GFP or mCherry alone did not disrupt axon morphology. We have also ruled out the possibility that the observed degeneration is caused by manipulating and imaging the worms. In mosaic animals, *tir1* expressing axons degenerate while non-*tir-1* expressing axons maintain a wild type morphology, even though they are in the same animal (Extended Data Figure 4). We conclude that expression of *tir-1* does induce axon degeneration.

**Author response table 1. sa2table1:** Motor axon degeneration is dependent on expression of *tir-1*, but not *mCherry*. *tir1(qd4)* animals were injected with the indicated expression constructs and the number of degenerating GABA motor axons was quantified. Expression of GFP or mCherry in motor axons is not sufficient to induce axon degeneration on their own. However, expression of untagged *tir-1* or GFP-tagged *tir-1* does induce axon degeneration.

Description	Expression Plasmid	% Damaged Axons
control	*unc-47p::GFP*	0
control	*unc-47p::mCherry*	0
untagged line 1	*unc-47p::tir-1*	9
untagged line 2	*unc-47p::tir-1*	6
untagged line 3	*unc-47p::tir-1*	11
untagged line 4	*unc-47p::tir-1*	17
GFP tagged line 1	*unc-47p::tir-1::GFP*	13
GFP tagged line 2	*unc-47p::tir-1::GFP*	31
GFP tagged line 3	*unc-47p::tir-1::GFP*	6

Thank you for bringing this to our attention. Wild type axons are less likely to regenerate when the middle segment has degenerated and are more likely to regenerate when the middle segment has not degenerated. We agree that based on this data alone, the mechanisms associated with protection of the middle fragment could promote regeneration in wild type animals or the mechanisms associated with degeneration of the middle fragment could inhibit axon regeneration. Because we cannot differentiate between the two possibilities, we have removed this section from the manuscript.

We thank the reviewer for their comments. We agree that the implications of our finding that TIR-1 regulates axon regeneration are large and are of significant interest to the field, particularly if TIR-1 and Sarm1 are functional orthologs.

Multiple lines of evidence indicate TIR-1 is indeed a functional ortholog of dSARM and Sarm1. The three genes have evolved through speciation and not duplication^9^, share extensive sequence homology^10^, domain conservation^1,10-12^ and overall structural identity^1-4,10,11,13^. TIR-1, dSarm and Sarm1 also share molecular interactions and functions. They interact with conserved MAPK signal transduction pathways^4,11,12,14-19^ and all three are NAD^+^ hydrolases^1-5,7,13^. In addition, the genes regulate a number of shared cellular processes, including the innate immune response^11,19-28^ and cell death^1,29,30^.

Our data adds the significant finding that, like dSarm and SARM1^31,32^, TIR-1 positively regulates injury induced axon degeneration. Specifically, we found (1) loss of TIR-1 suppresses degeneration of injured axons (Figure 3), (2) expression of a TIR-1 isoform that lacks its autoinhibitory N-terminus induces axon degeneration (Figure 4), and () TIR-1 interacts with DLK MAP Kinase signaling to regulate injuryinduced axon degeneration (Figure 6). In addition to regulating degeneration in response to injury, TIR-1, dSarm and SARM1 also regulate degeneration induced by disease, age and toxicity^18,33-37^.

The reviewer’s comment inspired us to specifically address the question of whether TIR-1 and human SARM1 might have orthologous function as cell-autonomous regulators of axon regeneration. To determine whether human Sarm1 (hSarm1) could function cell-autonomously to inhibit axon regeneration, we expressed low levels of hSARM1 in *C. elegans* GABA motor axons and asked whether it was sufficient to rescue the regeneration phenotype of *tir-1(qd4)* null animals. We found that cell-intrinsic expression of hSARM1 does rescue the motor axon regeneration phenotype (Figure 2F). Therefore, hSARM1 is capable of regulating axon regeneration within injured neurons, which is an additional line of evidence that hSARM1 and TIR-1 function are evolutionarily conserved. These results have been added to the manuscript at line 166.

Together, the sequence homology, structural conservation, and ability to regulate multiple shared functions, including degeneration and regeneration, of TIR-1, dSarm, and SARM1 strongly indicate they are functional orthologs. We therefore believe that our data significantly contribute to our understanding of both TIR-1/SARM1 function and the injury response.

We agree with the reviewer that finding only 20% of degeneration is suppressed in *tir-1(-)* animals is different from what might be expected given the extensive conservation of function between *tir-1*, *dSarm*, and *Sarm1*. However, directly comparing the magnitude of the effect that SARM1 has in inducing degeneration with the magnitude of the effect of TIR-1 is not possible, as these are different organisms, different neurons and different injury paradigms, all of which could influence the potential of an injured axon to degenerate. Indeed, many one-to-one orthologs regulate conserved processes in a complex manner that usually reflects a degree of plasticity in the adaptation of organisms to their environment. For example, the NADase activity of dSarm in *Drosophila* differs from that of its human, mouse or zebrafish SARM orthologs, such that dSarm generates an inverse ratio of cyclic ADPR to ADP-ribose compared to its orthologs^7^.

Here, in addition to the loss of function phenotype of *tir-1(-)* animals, we found a strong relationship between expression levels of TIR-1 and the degree of injury-induced axon degeneration. Increasing TIR-1 expression levels and activity causes profound and penetrant injury-induced degeneration (Figure 4A-E), while decreasing TIR-1 function reduces injury-induced degeneration (Figure 3G). These data, together with multiple lines of evidence that the sequence, structure, and functions of TIR-1 are conserved with dSarm and SARM1 (outlined in the previous response), indicate that TIR-1 is indeed a true ortholog of dSarm and SARM1 that positively regulates injury-induced degeneration.

As the reviewer notes, this raises the question of whether TIR-1 promotes axon degeneration similarly to dSarm and SARM1? We have now found that the modest but significant and penetrant reduction in degeneration in the absence of *tir-1* can be attributed, at least in part, to the following: (1) the presence of degeneration in the absence of *tir-1* function indicates additional mechanisms function in parallel to *tir-1* to regulate injury-induced degeneration in *C. elegans* (Figure 3G), (2) *tir-1* regulates degeneration with non-canonical NADase independent, potentially conserved mechanisms (Figure 6A and see following response) and (3) while distal axon fragments do not degenerate after a single axotomy, expression of an active *tir-1b* isoform strongly induces injury-dependent degeneration specifically in the distal fragment of singly axotomized axons. Because we found that *tir-1* is expressed in the non-regenerating axons and does promote degeneration of completely severed middle fragments, these data indicate *tir-1* function may be insufficiently activated or suppressed in specific cellular contexts (Figure 4A-E and see response to reviewer 3).

Knowing both the similarities and the differences in how TIR-1 and its SARM orthologs regulate their shared functions is critical to understanding the multiple ways in which regeneration and degeneration can be regulated, which is a prerequisite to learning how to manipulate the identified mechanisms in different contexts to repair and protect the nervous system.

We thank the reviewer for this suggestion. To determine whether TIR-1 regulates injury-induced degeneration through its NAD^+^ hydrolase activity, we CRISPR edited the key catalytic glutamate E788/E642^2,7,13^ (Figure 6A). Interestingly, the E788A/E642A mutation did not suppress injury-induced axon degeneration (Figure 6A). Since E788A is reported to render the TIR domain of *C. elegans* TIR-1 catalytically dead^4^, these data indicate that TIR-1 regulates axon degeneration independently of its NAD^+^ hydrolase activity. Our finding is consistent with recent results from the Freeman lab, which show mutation of the corresponding catalytic residue in dSarm (E1170A/E788A/E642A) only partially protects axons from degenerating^14^. Together, these results indicate that the ability to regulate degeneration independently of NAD^+^ hydrolysis may be conserved between TIR-1 and dSarm. What the additional NADase-independent mechanisms are and whether they are shared between TIR-1 and dSarm remain exciting open questions. These data have been added to the manuscript at Line 305 and in Figure 6A.

We also asked whether other components of the TIR domain are important for its ability to regulate axon degeneration. In addition to its key E788/E642 catalytic residue, TIR domain function requires a structure specific to catalytically active TIR domains called a SARM Specific (SS) loop (residues 622635), a BB loop (residues 594-605) thought to be important for intradomain interactions, and multiple residues important for protein multimerization and interaction with the autoinhibitory domain^1-3,7,13^. D627 is a conserved residue in the SS loops of *C. elegans*, human, and mouse TIR domains that is required for the NADase activity of SARM1 and for its ability to induce degeneration of injured DRGs, but not for its ability to multimerize^1,2^. Interestingly, we found that axon degeneration in *tir-1(D773A)* mutants was significantly disrupted compared to wild-type animals and was similar to that seen in *tir-1(qd4)* animals (Figure 6A). These data indicate the SS loop is important for TIR-1’s ability to promote degeneration of injured axons. Collectively with the findings that (1) mutation of the key catalytic glutamate E788A abolishes NADase activity in the TIR domain of TIR-1^4^ but not degeneration (Figure 6A), and (2) the SS-loop residue D627 is required for degeneration and promotes the NADase activity of SARM1 through relatively unknown mechanisms^1,2^, our data indicate that D773A suppresses degeneration by disrupting an as yet unidentified function of TIR-1. Notably, D773A is not a null mutation, since it did not disrupt the ability of TIR-1 to inhibit regeneration (Figure 6A). It follows that TIR-1 regulates axon regeneration and degeneration by different mechanisms, potentially by directly interacting with components of the UNC-43-NSY-1PMK-1 and DLK-1-PMK-3 MAP kinase pathways on either side of the injury. These data have been added to the manuscript at Line 316 and in Figure 6A.

Future experiments will determine whether the amount of available NAD^+^ regulates injury-induced degeneration of *C. elegans* motor neurons independently of TIR-1’s NADase activity. Because this has not yet been determined, we have carefully avoided defining the injury-induced degeneration as Wallerian degeneration and instead noted that “The degeneration has similar morphological and temporal features as Wallerian degeneration, where after a brief latent period, the middle axon segment beads, fragments, and is eventually cleared (Figure 3B).” (Line 144)

As the reviewer points out, in non-injured chronically degenerating axons that overexpress the *tir-1b* isoform, our data agrees with previously published work^15^: DLK-1 functions downstream of TIR-1. As detailed below, our data also supports a bimodal relationship between TIR1 and DLK-1 that is dependent on the level of TIR-1 expression, the presence of the N-terminal autoinhibitory domain of TIR-1, and a traumatic injury event. While we believe this is a very interesting avenue of further investigation, we have removed this experiment from the manuscript to focus specifically on how the endogenous *tir-1* gene regulates injury-induced degeneration.

A longstanding point of confusion in the Sarm1 literature is how MAPK signaling interacts with Sarm1. Separate studies found that MAPK functions downstream of Sarm1 to regulate degeneration of crushed retinal ganglion cells and upstream of Sarm1^15,17^ to regulate degeneration of cultured mammalian dorsal root ganglia neurons and *Drosophila* motoneurons. These data indicate that the complex relationship between SARM1 and MAPK signaling may differ according to cell-type, assay, and species.

We found that TIR-1 interacts with DLK-1 MAPK signaling to regulate degeneration of injured motor axons. If, as the reviewer suggests, *tir-1* promotes axon degeneration by inhibiting *dlk-1* and *pmk-3*, one would expect that loss of *dlk-1* and *pmk-3* would increase axon degeneration relative to wild type. In the initial experiments, determining the nature of the relationship between *tir-1* and *dlk-1* was made difficult by a ceiling effect in the assay. Since almost all wild-type axons degenerate, it is difficult to determine whether a given mutant displays significantly more degeneration compared to wild type animals. This ceiling effect confounds interpretation of whether *dlk-1* and *pmk-3* suppress degeneration and if so, whether *tir-1* functions upstream of *dlk-1* and *pmk-3* or whether *tir-1* partially blocks *dlk-1* and *pmk-3* signaling.

To determine whether *dlk-1* does inhibit axon degeneration and how it interacts with *tir-1*, we repeated the epistasis analysis using a gain of function allele of *dlk-1*. We found that overexpressing *dlk-1* suppressed axon degeneration. This suppression is not likely to be a dominant negative effect since the opposite *dlk-1* loss of function mutation does not suppress degeneration. We also found that *dlk-1* induced protection was partially dependent on *tir-1* function, indicating *dlk-1* is capable of inhibiting degeneration of injured motor neurons by regulating *tir-1* function and by regulating another as yet unknown mechanism of degeneration. Therefore, together with the multifaceted relationship between Sarm1 and MAPK signaling, our data indicate the highly conserved TIR-1, dSarm and Sarm1 proteins regulate injury-induced degeneration in coordination with shared MAPK signaling components. However, the precise interactions between TIR-1/Sarm1/dSarm, DLK1/MAPK/Wnd signaling, and degeneration are complex and likely depend on cell type, form of injury, and animal system. Identifying the similarities and differences between the various interactions is important to understanding how these complex genes regulate the injury response. We have added this data at Line 348 and included it in Figure 6.

We have modified this section to clarify the results. Since *tir-1*, *nsy-1*, and *pmk-1* null mutants all inhibit axon regeneration to the same extent, and because the amount of regeneration in any pairwise combination of mutants is not additive or multiplicative compared to the single mutants, our data indicate that the three genes function in the same pathway to regulate axon regeneration. Line 274 and Figure 5D.

Our data indicate that *dlk-1* function is required for axon regeneration in *tir-1*(-) and *pmk-3(-)* animals (Figure 5B)*.* However, *dlk-1* function is also absolutely required for regeneration of otherwise wildtype GABA motor axons^38,39^. Physical interpretations of genetic interactions with genes that are essential for a given function can be misleading if that function requires the activity of multiple distinctly regulated processes. In this experiment, since DLK-1 function is absolutely required for axon regeneration, we cannot distinguish whether TIR-1 signaling regulates the activity of DLK-1 or whether DLK-1 regulates an essential component of axon regeneration independently of TIR-1 signaling. However, since the enhanced regeneration in *tir-1(qd4)* mutants requires *dlk-1,* the genetic interaction does reveal that TIR-1 does not function downstream of, or redundantly with, DLK-1. Finally, the finding that loss of *nsy-1* or *pmk-1* causes the opposite regeneration phenotype to loss of *dlk-1* or *pmk-3* indicates the two sets of genes must not act redundantly. Together, our data indicate that *tir-1* functions with *nsy-1* and *pmk-1* to inhibit axon regeneration, which is dependent on *dlk-1*. Line 249 and Figure 5B.

In the figures listed above, the amount of WT regeneration differs because they represent the frequency of regeneration after different types of injury. The frequency of regeneration in wild type animals is consistent within each particular type of assay.

We control against day-to-day and instrument to instrument variability in experiments by performing same day controls on the same axotomy rig, as is standard for *C. elegans* axon regeneration experiments. In addition to referring to the methods paper that describes these practices^40^, we have added a description of the axotomy protocol in the Materials and methods section.

Thank you for your positive comments.

We thank the reviewer for their positive comments about our regeneration data and its significance. We believe that the finding that TIR-1 regulates injury-induced degeneration is also significant because it demonstrates (1) that TIR-1 is a functional ortholog of the other SARMs (see response to reviewer two on page 4) and (2) that the mechanism by which TIR-1 regulates regeneration is distinct from how it regulates degeneration. We have revised the manuscript accordingly.

We also agree that understanding how degeneration is differentially regulated in *C. elegans* is a very interesting avenue of investigation. Our data contributes to our understanding of two exciting questions in this regard:

1) Why does TIR-1 only induce degeneration after double axotomy and not after a single axotomy?

We hypothesized that perhaps distal fragments do not degenerate after single axotomy because TIR-1 is not sufficiently activated by this type injury in *C. elegans*. TIR-1 and Sarm1 functions are autoinhibited by their N-terminal domains^1,12,13,32,45-48^. We found that when an endogenous isoform of *tir-1* that lacks its N-terminal autoinhibitory domain is expressed at low levels, a single axotomy induces degeneration of distal stumps (Figure 4A-E). These data indicate that in wild type *C. elegans*, axons that have been injured once do not degenerate because the injury response was of insufficient magnitude or identity to activate TIR-1. In contrast, double axotomy induces an enhanced injury response of sufficient magnitude or identity to activate TIR-1 or overcome its suppression. In future experiments that are beyond the scope of the current manuscript we will investigate whether and how TIR-1 is differentially activated after single and double axotomy. We have added this data to the manuscript at Line 185 and Figure 4.

2) Why does loss of TIR-1 function not completely suppress axon degeneration?

Our finding that axons are only partially protected from degenerating in the absence of *tir-1* (Figure 3G) reveals that multiple mechanisms regulate degeneration of injured motor axons in *C. elegans*. We asked whether *ced-1,* a homolog of *Draper* and *Megf10*, also contributes to the degeneration phenotype, since it functions in the hypodermis and muscle cells of *C. elegans* to remove axonal debris after injury in mechanosensory neurons^49,50^. Here, doubly axotomized motor axons in *ced-1* loss of function mutant animals displayed wild type levels of axon degeneration (Extended Data Figure 3). Moreover, animals with loss of function mutations in both *ced-1* and *tir-1* phenocopied the degeneration observed in *tir-1* animals. These results demonstrate that middle fragment degeneration after double injury does not depend on *ced-1.* Together with our finding that injury-induced degeneration is not completely dependent on *tir1* (Figure 3G), these data indicate *C. elegans* motor axons have additional as yet unidentified regulators of injury-induced motor axon degeneration. We have added this data to the manuscript at Line 173 and Extended Data Figure 3.

We believe it will be difficult to definitively determine whether injuring an axon in two places is more or less artificial than injuring an axon precisely at the midline with a laser. However, since double axotomy induces degeneration of severed axons in *C. elegans*, we believe that the resulting injury response more accurately reflects that of other systems. This led us to ask whether the frequency of axon regeneration differs in the presence or absence of degeneration. The result is that, after double injury, axons regenerate less frequently when the middle fragment had degenerated compared to when the fragment remains intact. This data reveals a difference in the molecular mechanisms that regulate regeneration in the presence and absence of degeneration.

We agree with the reviewer that it is difficult to interpret this data and because we do not know how the interaction between regeneration and degeneration is regulated, we have removed figure 3H and mention of the relationship between regeneration and middle fragment degeneration from the manuscript.

References

Summers, D. W., Gibson, D. A., DiAntonio, A. & Milbrandt, J. SARM1-specific motifs in the TIR domain enable NAD+ loss and regulate injury-induced SARM1 activation. *Proc Natl Acad Sci U S A* 113, E6271-E6280, doi:10.1073/pnas.1601506113 (2016).Loring, H. S., Icso, J. D., Nemmara, V. V. & Thompson, P. R. Initial Kinetic Characterization of Sterile Α and Toll/Interleukin Receptor Motif-Containing Protein 1. *Biochemistry* 59, 933-942, doi:10.1021/acs.biochem.9b01078 (2020).Loring, H. S. *et al.* A phase transition enhances the catalytic activity of SARM1, an NAD(+) glycohydrolase involved in neurodegeneration. *ELife* 10, doi:10.7554/*eLife*.66694 (2021).Peterson, N. D. *et al.* Pathogen infection and cholesterol deficiency activate the *C. elegans* p38 immune pathway through a TIR-1/SARM1 phase transition. *ELife* 11, doi:10.7554/*eLife*.74206 (2022).Gerdts, J., Summers, D. W., Milbrandt, J. & DiAntonio, A. Axon Self-Destruction: New Links among SARM1, MAPKs, and NAD+ Metabolism. *Neuron* 89, 449-460, doi:10.1016/j.neuron.2015.12.023 (2016).Horsefield, S. *et al.* NAD^+^ cleavage activity by animal and plant TIR domains in cell death pathways. *Science* 365, 793-799, doi:doi:10.1126/science.aax1911 (2019).Essuman, K. *et al.* The SARM1 Toll/Interleukin-1 Receptor Domain Possesses Intrinsic NAD(+) Cleavage Activity that Promotes Pathological Axonal Degeneration. *Neuron* 93, 1334-1343 e1335, doi:10.1016/j.neuron.2017.02.022 (2017).Essuman, K. *et al.* TIR Domain Proteins Are an Ancient Family of NAD(+)-Consuming Enzymes. *Curr Biol* 28, 421-430 e424, doi:10.1016/j.cub.2017.12.024 (2018).Thompson, J. D., Gibson, T. J. & Higgins, D. G. Multiple sequence alignment using ClustalW and ClustalX. *Curr Protoc Bioinformatics* Chapter 2, Unit 2 3, doi:10.1002/0471250953.bi0203s00 (2002).Mink, M., Fogelgren, B., Olszewski, K., Maroy, P. & Csiszar, K. A novel human gene (SARM) at chromosome 17q11 encodes a protein with a SAM motif and structural similarity to Armadillo/β-catenin that is conserved in mouse, *Drosophila*, and *Caenorhabditis elegans*. *Genomics* 74, 234-244, doi:10.1006/geno.2001.6548 (2001).Liberati, N. T. *et al.* Requirement for a conserved Toll/interleukin-1 resistance domain protein in the *Caenorhabditis elegans* immune response. *Proc Natl Acad Sci U S A* 101, 6593-6598, doi:10.1073/pnas.0308625101 (2004).Chuang, C. F. & Bargmann, C. I. A Toll-interleukin 1 repeat protein at the synapse specifies asymmetric odorant receptor expression via ASK1 MAPKKK signaling. *Genes Dev* 19, 270-281, doi:10.1101/gad.1276505 (2005).Horsefield, S. *et al.* NAD+ cleavage activity by animal and plant TIR domains in cell death pathways. *Science* 365, 793-799, doi:10.1126/science.aax1911 (2019).Hsu, J. M. *et al.* Injury-Induced Inhibition of Bystander Neurons Requires dSarm and Signaling from Glia. *Neuron* 109, 473-487 e475, doi:10.1016/j.neuron.2020.11.012 (2021).Yang, J. *et al.* Pathological axonal death through a MAPK cascade that triggers a local energy deficit. *Cell* 160, 161-176, doi:10.1016/j.cell.2014.11.053 (2015).Miller, B. R. *et al.* A dual leucine kinase-dependent axon self-destruction program promotes Wallerian degeneration. *Nat Neurosci* 12, 387-389, doi:10.1038/nn.2290 (2009).Walker, L. J. *et al.* MAPK signaling promotes axonal degeneration by speeding the turnover of the axonal maintenance factor NMNAT2. *ELife* 6, doi:10.7554/*eLife*.22540 (2017).Ding, C., Wu, Y., Dabas, H. & Hammarlund, M. Activation of the CaMKII-Sarm1-ASK1-p38 MAP kinase pathway protects against axon degeneration caused by loss of mitochondria. *ELife* 11, doi:10.7554/*eLife*.73557 (2022).Couillault, C. *et al.* TLR-independent control of innate immunity in *Caenorhabditis elegans* by the TIR domain adaptor protein TIR-1, an ortholog of human SARM. *Nat Immunol* 5, 488-494, doi:10.1038/ni1060 (2004).Hou, Y. J. *et al.* SARM is required for neuronal injury and cytokine production in response to central nervous system viral infection. *J Immunol* 191, 875-883, doi:10.4049/jimmunol.1300374 (2013).Gurtler, C. *et al.* SARM regulates CCL5 production in macrophages by promoting the recruitment of transcription factors and RNA polymerase II to the Ccl5 promoter. *J Immunol* 192, 4821-4832, doi:10.4049/jimmunol.1302980 (2014).Carty, M. *et al.* Cell Survival and Cytokine Release after Inflammasome Activation Is Regulated by the Toll-IL-1R Protein SARM. *Immunity* 50, 1412-1424 e1416, doi:10.1016/j.immuni.2019.04.005 (2019).Carty, M. *et al.* The human adaptor SARM negatively regulates adaptor protein TRIF-dependent Toll-like receptor signaling. *Nat Immunol* 7, 1074-1081, doi:10.1038/ni1382 (2006).Akhouayri, I., Turc, C., Royet, J. & Charroux, B. Toll-8/Tollo negatively regulates antimicrobial response in the *Drosophila* respiratory epithelium. *PLoS Pathog* 7, e1002319, doi:10.1371/journal.ppat.1002319 (2011).Baral, P. & Utaisincharoen, P. Sterile-α- and armadillo motif-containing protein inhibits the TRIF-dependent downregulation of signal regulatory protein α to interfere with intracellular bacterial elimination in Burkholderia pseudomallei-infected mouse macrophages. *Infect Immun* 81, 3463-3471, doi:10.1128/IAI.00519-13 (2013).Dinić, M. *et al.* Probiotic-mediated p38 MAPK immune signaling prolongs the survival of *Caenorhabditis elegans* exposed to pathogenic bacteria. *Scientific Reports* 11, doi:10.1038/s41598021-00698-5 (2021).Kurz, C. L., Shapira, M., Chen, K., Baillie, D. L. & Tan, M. W. *Caenorhabditis elegans pgp-5* is involved in resistance to bacterial infection and heavy metal and its regulation requires TIR-1 and a p38 map kinase cascade. *Biochemical and Biophysical Research Communications* 363, 438-443, doi:10.1016/j.bbrc.2007.08.190 (2007).Pudla, M., Limposuwan, K. & Utaisincharoen, P. Burkholderia pseudomallei-induced expression of a negative regulator, sterile-α and Armadillo motif-containing protein, in mouse macrophages: a possible mechanism for suppression of the MyD88-independent pathway. *Infect Immun* 79, 29212927, doi:10.1128/IAI.01254-10 (2011).Hayakawa, T. *et al.* Regulation of anoxic death in *Caenorhabditis elegans* by mammalian apoptosis signal-regulating kinase (ASK) family proteins. *Genetics* 187, 785-792, doi:10.1534/genetics.110.124883 (2011).Kim, W., Underwood, R. S., Greenwald, I. & Shaye, D. D. OrthoList 2: A New Comparative Genomic Analysis of Human and *Caenorhabditis elegans* Genes. *Genetics* 210, 445-461, doi:10.1534/genetics.118.301307 (2018).Osterloh, J. M. *et al.* dSarm/Sarm1 is required for activation of an injury-induced axon death pathway. *Science* 337, 481-484, doi:10.1126/science.1223899 (2012).Gerdts, J., Summers, D. W., Sasaki, Y., DiAntonio, A. & Milbrandt, J. Sarm1-mediated axon degeneration requires both SAM and TIR interactions. *J Neurosci* 33, 13569-13580, doi:10.1523/JNEUROSCI.1197-13.2013 (2013).Lezi, E. *et al.* An Antimicrobial Peptide and Its Neuronal Receptor Regulate Dendrite Degeneration in Aging and Infection. *Neuron* 97, 125-138 e125, doi:10.1016/j.neuron.2017.12.001 (2018).Veriepe, J., Fossouo, L. & Parker, J. A. Neurodegeneration in *C. elegans* models of ALS requires TIR1/Sarm1 immune pathway activation in neurons. *Nat Commun* 6, 7319, doi:10.1038/ncomms8319 (2015).Henninger, N. *et al.* Attenuated traumatic axonal injury and improved functional outcome after traumatic brain injury in mice lacking Sarm1. *Brain* 139, doi:10.1093/brain/aww001 (2016).Massoll, C., Mando, W. & Chintala, S. K. Excitotoxicity upregulates SARM1 protein expression and promotes Wallerian-like degeneration of retinal ganglion cells and their axons. *Invest Ophthalmol Vis Sci* 54, 2771-2780, doi:10.1167/iovs.12-10973 (2013).Geisler, S. *et al.* Prevention of vincristine-induced peripheral neuropathy by genetic deletion of SARM1 in mice. *Brain* 139, 3092-3108, doi:10.1093/brain/aww251 (2016).Hammarlund, M., Nix, P., Hauth, L., Jorgensen, E. M. & Bastiani, M. Axon regeneration requires a conserved MAP kinase pathway. *Science* 323, 802-806, doi:10.1126/science.1165527 (2009).Yan, D., Wu, Z., Chisholm, A. D. & Jin, Y. The DLK-1 kinase promotes mRNA stability and local translation in *C. elegans* synapses and axon regeneration. *Cell* 138, 1005-1018, doi:10.1016/j.cell.2009.06.023 (2009).Byrne, A. B., Edwards, T. J. & Hammarlund, M. in vivo laser axotomy in *C. elegans*. *J Vis Exp*, doi:10.3791/2707 (2011).Wang, Q. *et al.* Sarm1/Myd88-5 Regulates Neuronal Intrinsic Immune Response to Traumatic Axonal Injuries. *Cell Rep* 23, 716-724, doi:10.1016/j.celrep.2018.03.071 (2018).Neumann, B., Nguyen, K. C., Hall, D. H., Ben-Yakar, A. & Hilliard, M. A. Axonal regeneration proceeds through specific axonal fusion in transected *C. elegans* neurons. *Dev Dyn* 240, 1365-1372, doi:10.1002/dvdy.22606 (2011).Sasaki, Y., Nakagawa, T., Mao, X., DiAntonio, A. & Milbrandt, J. NMNAT1 inhibits axon degeneration via blockade of SARM1-mediated NAD(+) depletion. *ELife* 5, doi:10.7554/*eLife*.19749 (2016).Kim, K. W. *et al.* Expanded genetic screening in *Caenorhabditis elegans* identifies new regulators and an inhibitory role for NAD(+) in axon regeneration. *ELife* 7, doi:10.7554/*eLife*.39756 (2018).Shen, C. *et al.* Multiple domain interfaces mediate SARM1 autoinhibition. *Proc Natl Acad Sci U S A* 118, doi:10.1073/pnas.2023151118 (2021).Jiang, Y. *et al.* The NAD(+)-mediated self-inhibition mechanism of pro-neurodegenerative SARM1. *Nature* 588, 658-663, doi:10.1038/s41586-020-2862-z (2020).Figley, M. D. *et al.* SARM1 is a metabolic sensor activated by an increased NMN/NAD(+) ratio to trigger axon degeneration. *Neuron* 109, 1118-1136 e1111, doi:10.1016/j.neuron.2021.02.009 (2021).Sporny, M. *et al.* Structural basis for SARM1 inhibition and activation under energetic stress. *ELife* 9, doi:10.7554/*eLife*.62021 (2020).Nichols, A. L. A. *et al.* The Apoptotic Engulfment Machinery Regulates Axonal Degeneration in *C. elegans* Neurons. *Cell Rep* 14, 1673-1683, doi:10.1016/j.celrep.2016.01.050 (2016).Chiu, H. *et al.* Engulfing cells promote neuronal regeneration and remove neuronal debris through distinct biochemical functions of CED-1. *Nat Commun* 9, 4842, doi:10.1038/s41467-018-07291-x (2018).

[Editors’ note: what follows is the authors’ response to the second round of review.]

Essential revisions:1. While a subset of experiments addressing degeneration are performed after the single axotomy, others are performed after a double cut. It is unclear whether the mechanisms underlying degeneration in these different experimental paradigms is identical or different. Specifically, the authors should compare the role of the MAPK components for degeneration between the single and double-cut protocols.

In response to the reviewers’ comments, we have added significantly to our finding that DLK-1 and PMK-3 MAP kinase components regulate injury induced axon degeneration in wild type animals. Further investigation of double cut axons has revealed that DLK-1 and PMK-3 inhibit wild type degeneration of middle axon fragments that are severed from both the cell body and synaptic partner (detailed below in Essential revision 2). Whether the DLK-1 MAP kinase pathway also inhibits wild type degeneration of distal fragments after a single injury is an interesting question. Previously published work demonstrates the distal fragments of singly injured axons do not readily degenerate in wild type or *tir-1(-)* backgrounds 24 hours after injury (Nichols et al., 2016; Yanik et al., 2004). Here, we asked whether that inability to degenerate is determined by DLK-1 by investigating whether the severed distal fragments of *dlk-1(-)* or *dlk-1(oe)* axons degenerate after a single injury. We found no significant difference in the number of axons that degenerate between either of the *dlk-1* mutants and wild type animals, indicating DLK-1 does not promote or inhibit degeneration of distal fragments in this injury model (Figure 7—figure supplement 2). Very rarely, a severed distal fragment was absent from wild type or *dlk-1(oe)* animals; however, whether or not this represents an active form of degeneration, it is not regulated by DLK-1 MAPK signaling. Similarly, we did not observe distal fragment degeneration in singly injured *pmk-3(-)*, *nsy-1(-)*, or *pmk-1(-)* axons, indicating these MAP kinases do not inhibit degeneration of wild type distal axon fragments after a single injury.

In summary, our analysis of the role of DLK MAPK in *C. elegans* across diverse models of degeneration reveals: (1) DLK-1 inhibits degeneration of middle axon fragments that are severed from both the proximal and distal axon fragments by a double injury (See response to Essential revision 2 and Figure 7B,C), and (2) DLK-1 does not regulate degeneration of distal axon fragments after a single injury (Figure 7—figure supplement 2). We conclude that degeneration of *C. elegans* motor axons is differentially regulated by DLK-1, according to type of insult.

The dichotomous role of DLK-1 in axon degeneration agrees with the duality of MAPK signaling in established models of injury-induced degeneration. For example, Wallenda (Wnd), the DLK-1 homolog in *Drosophila melanogaster*, inhibits axon degeneration in motor axons that have undergone a conditioning lesion (Xiong & Collins, 2012). However, Wallenda also promotes degeneration in singly severed olfactory receptor axons and mammalian DLK promotes degeneration of severed dorsal root ganglion axons (Ghosh et al., 2011; Miller et al., 2009). While it is unlikely that the small amount of time between axotomies in the double injury model induces a conditioning lesion; the comparison demonstrates that Wnd and DLK1 activity is differentially determined by specific types of injury and by cell type. In relation to SARM signaling, separate studies have found that MAPK signaling functions downstream of SARM1 to regulate degeneration of crushed retinal ganglion cells, and upstream of SARM1 to regulate degeneration of cultured mammalian dorsal root ganglia neurons as well as *Drosophila* motoneurons (Summers et al., 2018; Walker et al., 2017; Yang et al., 2015). Identifying the similarities and differences between the various interactions across species and types of injury will be important to understanding how these complex genes regulate the injury response.

These data have been added to the manuscript as Figure 7—figure supplement 2 and are described at Line 424.

2. The role of DLK-1/TIR-1 in axon degeneration needs to be clarified with further experiments. The data do not always support the conclusions and in some cases, the two contradict each other – please see points 8-10 from Reviewer 2.

We performed additional experiments to clarify the role of TIR-1 and DLK-1 in injury-induced axon degeneration. We previously found that loss of *tir-1* function reduces the degeneration frequency of middle axon fragments that are severed from both the proximal and distal fragments (Figure 3G). Therefore, TIR-1 promotes degeneration of severed GABA motor axon fragments. Our analysis of loss of function mutations in *dlk-1* and its canonical downstream MAPK, *pmk-3*, revealed that both DLK-1 and PMK-3 inhibit degeneration of severed middle axon fragments. Specifically, we found that loss of *dlk-1* or *pmk-3* function alone did not significantly affect degeneration of severed middle fragments compared to wild type animals; however, loss of *dlk-1* or *pmk-3* function increased the frequency of middle fragment degeneration in *tir-1(qd4)* mutants (Figure 7B). Therefore, wild type DLK-1 and PMK-3 inhibit axon degeneration, at least in the absence of *tir-1(qd4)*. As outlined in the above response to Essential revision 1, the conclusion that DLK-1 inhibits axon degeneration is analogous to the finding that Wallenda, its *Drosophila* homolog, also inhibits degeneration of injured motor neurons that have undergone a conditioning lesion (Xiong & Collins, 2012). However, it differs from the finding that Wallenda promotes degeneration in singly severed olfactory receptor axons and that DLK promotes degeneration of severed dorsal root ganglion axons (Ghosh et al., 2011; Miller et al., 2009).

In response to the reviewers’ comments, we further investigated the epistatic relationship between *tir-1* and *dlk-1* using animals that overexpress *dlk-1* (*dlk-1(oe)*). We first outcrossed the *tir-1(qd4*) mutation from *dlk-1(oe); tir-1(qd4)* animals because we found that the integrated high copy *dlk-1* array was being silenced over successive generations. Therefore, rebuilding the strains allowed us to compare the phenotypes of animals expressing the same transgenic allele, which originated from the same strain and had not been passaged for multiple generations. We verified that the newly isolated *dlk-1(oe)* and *dlk-1(oe)*; *tir-1(qd4)* animals contained and expressed similar amounts of mRNA from the integrated *dlk-1* transgene using qRT-PCR (Figure 7—figure supplement 1). Axons in *tir-1(-)* and *dlk-1(oe)* animals degenerated significantly less frequently than in wild type animals (Figure 7B,C). These data reinforce that *tir-1* promotes degeneration and *dlk-1* inhibits degeneration. The suppression of degeneration in doubly injured *dlk-1(oe)* axons was not dependent on *tir-1* function (Figure 7B). These data could be interpreted in at least two ways. First, *dlk-1* inhibits degeneration downstream of *tir-1* and therefore removing *tir-1* function has no consequence on a gene that is already activated. Alternatively, *dlk-1* inhibits degeneration at least in part by suppressing the function of *tir-1*. If so, loss of *tir-1* function does not affect the *dlk-1(oe)* phenotype because *tir-1* has already been inhibited.

We next asked whether blocking *dlk-1* signaling in the *dlk-1(oe)* strain would reveal the nature of the genetic relationship between *dlk-1* signaling and *tir-1(qd4)*. We previously found that loss of function mutations in *dlk-1* and its canonical downstream MAPK *pmk-3* cause similar degeneration phenotypes, both in the presence and absence of *tir-1* function (Figure 7B). The phenotypic similarity prompts the hypothesis that *pmk-3* functions downstream of *dlk-1* to inhibit axon degeneration. We asked whether the inhibition of degeneration observed in *dlk1(oe)* animals does depend on *pmk-3* function by quantifying degeneration in *dlk-1(oe); pmk3(-)* animals. Loss of *pmk-3* function rescued degeneration in animals that overexpress *dlk-1* (Figure 7B); therefore, *dlk-1* functions with its downstream MAPK *pmk-3* to inhibit degeneration.

With the ability to block *dlk-1* function in *dlk-1(oe)* animals, we could further investigate how *dlk-1*–*pmk-3* signaling interacts with *tir-1*. Loss of *pmk-3* function rescued degeneration in *dlk-1(oe); tir-1(-)* animals, indicating DLK-1–PMK-3 signaling does not inhibit degeneration by inhibiting TIR-1 (Figure 7B). Instead, DLK-1–PMK-3 signaling inhibits an as yet unknown effector of degeneration that can function even in the absence of TIR-1. Given the ceiling effect in the assay, we cannot rule out the possibility that *pmk-3* also functions downstream of another inhibitor of degeneration; however, we can conclude that DLK-1-PMK-3 signaling is genetically epistatic to TIR-1.

These data have been added to the manuscript as Figure 7 and Figure 7—figure supplement 1, and is described in Lines 413-427.

3. An NAD-independent function for TIR-1 in promoting degeneration is of great interest but this requires further substantiation, particularly in the context of being used in the somewhat unusual double-cut injury model. Does the expression of NADase-dead activated TIR-1 in the single cut model and overexpression of TIR-1 in the spontaneous degeneration model promote degeneration?

We found that in wild type axons, injury-induced regeneration and degeneration is regulated by non-canonical NADase-independent TIR-1 activity after a double injury (Figures 5A and 6A). In response to the reviewers’ comment, we asked whether the NADase activity of TIR-1 was required for its ability to promote degeneration in two additional models.

A) Spontaneous axon degeneration induced by overexpression of TIR-1b. We first asked whether overexpressing TIR-1b is capable of inducing NADase dependent axon degeneration in the absence of an acute injury. To do so, we mutated E788A/E642A and D773A/D627A in the TIR-1b expression construct, injected each into *tir-1(qd4)* animals, and isolated multiple independent transgenic strains. Expressing TIR-1b(E788A/E642A), which encodes a mutation that abolishes the NADase activity of TIR-1 and SARM1, did not induce spontaneous axon degeneration (Figure 6B) (Essuman et al., 2017; Horsefield et al., 2019; Loring et al., 2021; Peterson et al.; Summers et al., 2016). Nor did expression of TIR-1b(D773A/D627A), which encodes a mutation that disrupts multiple aspects of TIR-1 function, including NADase activity (Loring et al., 2021; Summers et al., 2016). The absence of spontaneous degeneration in the presence of TIR-1b(D773A) and TIR1b(E788A) is not attributed to differential expression of the respective transgenes compared to overexpression of the wild type *tir-1b* transgene, which was determined by quantifying expression of the same fluorescent tag on each protein (Figure 6C). However, because a small but statistically insignificant number of damaged axons were observed, we cannot rule out the possibility that greater TIR-1b(D773A) or TIR-1b(E788A) expression might induce NADase independent degeneration. Together with our previous results, we find that D773/D627 is required for both injury induced and spontaneous degeneration, while E788/E642 is only required for spontaneous degeneration. These data indicate D773A disrupts multiple aspects of TIR-1 function, and the NADase activity of TIR-1 is required to induce spontaneous degeneration in uninjured axons.

We note that these results do not substantiate or confound the finding that TIR-1 regulates middle fragment degeneration independently of its NADase activity. Rather, our data strongly support the conclusion that TIR-1 is capable of actively inducing degeneration of middle axon fragments without its enzymatic activity. The strength of this conclusion is owed to: (1) The nature of the E788A mutation. E788A is a CRISPR generated mutation of the endogenous *tir-1* locus, and therefore is a highly specific and system-wide mutation. (2) Previously published biochemical evidence that E788A does abolish the NADase activity of TIR-1 (Peterson et al.). And (3) The finding that middle fragment degeneration in *tir-1* mutants can be genetically suppressed by the loss of *dlk1* or *pmk-3* function (Figure 7B) indicates that TIR-1 regulates degeneration of severed middle fragments by inhibiting DLK-1–PMK-3 MAPK signaling. The ability to drive degeneration independently of NADase activity is potentially unique to the TIR-1 homolog, as the enzymatic activity of dSarm and SARM1 is required for their ability to regulate injury-induced degeneration (Brace et al., 2022; Herrmann et al., 2022; Hsu et al., 2021). However, dSARM has been found to regulate axonal transport after injury in an NADase independent manner demonstrating a key role for dSARM not requiring its NADase function (Hsu et al., 2021).

B) Degeneration of the distal axon fragment after a single injury in neurons that overexpress TIR-1b. In response to the reviewers’ comments, we also asked whether specifically disrupting the NADase activity of TIR-1b would block its ability to induce distal axon fragment degeneration after a single injury. We found that the distal stumps of axons that have been severed once do not degenerate significantly more frequently in axons that overexpress the *tir-1(E788A)* transgene compared to wild type axons. The significance of this data is diminished because the experiment combines injury and overexpression of *tir-1*, which we have found are regulated by different mechanisms.

Together, our finding that TIR-1 promotes degeneration independently of its NADase activity in wild type GABA motor axons that have been severed twice, along with the finding that TIR1 is capable of inhibiting degeneration of PVQ interneurons in the absence of mitochondria (Ding et al., 2022), and the inability of TIR-1 to promote spontaneous motor axon degeneration in the absence of its NADase activity, indicate TIR-1 regulates degeneration using multiple context specific mechanisms, at least two of which differ from how dSarm and SARM1 have been shown to regulate degeneration.

These data have been added to the manuscript as Figure 6 and described in Lines 389-404.

4. As you are no doubt aware, Marc Hammarlund recently showed that TIR-1 inhibits axon degeneration (albeit in a different neuron type) in contrast to the findings described here. This work is referenced only briefly; the contradictory findings and possible underlying reasons should be discussed in more detail.

We have included a more detailed description of the contrast between the protective and destructive functions of TIR-1: The Hammarlund lab found that TIR-1 can inhibit spontaneous axon degeneration of PVQ interneurons in mutant *ric-7(-)* animals (Ding et al., 2022), which lack mitochondria (Rawson et al., 2014). Together with our finding that TIR-1 promotes injury-induced degeneration of wild type GABA motor axons, along with previous findings that TIR-1 is required for degeneration in a *C.elegans* model of ALS, functions in the epidermis to promote age-dependent degeneration of PVD dendrites, and induces spontaneous degeneration when forced to multimerize, the data indicate endogenous TIR-1 differentially regulates degeneration in *C. elegans* in response to specific insults, contexts and potentially cell types (Ding et al., 2022; Lezi et al., 2018; Loring et al., 2021; Veriepe et al., 2015). Questions raised specifically from a comparison of the opposite roles of TIR-1 in response to injury and absence of mitochondria include whether a specific injury signal determines that TIR-1 takes on a destructive role and whether TIR-1 can protect multiple types of neurons against the spontaneous degeneration that is induced by the absence of mitochondria. Looking downstream, we found the map kinase *pmk-3* is epistatic to *tir-1* and inhibits injury-induced axon degeneration. Interestingly, *pmk-3* was also found to inhibit spontaneous degeneration in the absence of mitochondria, suggesting the difference between promotion and suppression of degeneration may ultimately be attributed to differential regulation of PMK-3 in these two contexts.

Reviewer #1 (Recommendations for the authors):Czech and colleagues present work showing that TIR-1 inhibits axon regeneration in *C. elegans* while also promoting degeneration in response to a double axon injury. They have adequately addressed the questions raised in the first round of review, and I feel that their work will attract broad interest in the field. Yet as they have added new and fascinating data, new questions necessarily arise. Below are several points that I feel the authors should clarify:

Thank you for your comments and enthusiasm for our findings.

- Regeneration is initially assayed after a single injury, for example in Figure 1 where the original tir-1 qd4 phenotype is shown. But by the time pathway analysis is performed in Figure 5 (showing genetic interactions with the MAPK pathway), regeneration is now characterized after double injury. The authors make no attempt to reconcile this very strange switch in paradigm. Was it simply out of convenience that these were the same animals assayed for injury-induced degeneration after double-cut? Regardless, the authors should show that the regenerative response (timing, magnitude, genetic dependences) are the same (or different?) between single and double cuts.

We have clarified the text to highlight the rationale for using the double injury model to investigate how TIR-1 regulates axon regeneration. We used the double injury model because we found that a double injury induces both degeneration and regeneration of wild type motor axons in the 24 hours following injury. In contrast, the distal segments of axons that have been severed once remain largely intact (Nichols et al., 2016; Yanik et al., 2004). The ability to observe both processes using one injury model is essential for comparing how TIR-1 regulates axon regeneration and degeneration within the same injured wild type axon and in the context of the same injury response.

We have added this clarification to the manuscript at Line 239-249.

We agree that it is also interesting to determine whether the regenerative responses are the same or different between axons that have been cut once or twice. We compared the timing, magnitude, genetic dependencies, and requirement for the NADase activity of TIR-1 between the two injury models. Each set of experiments is described below.

Magnitude and timing: Significantly less axons regenerate when they are injured twice, compared to when they are injured once. For example, in Figure 1D, 60.53% of singly cut wild type axons regenerate in the 24hrs following injury, compared to Figure 5B, in which 37.38% of doubly cut wild type axons regenerate in the same amount of time. Loss of *tir-1(-)* significantly rescues regeneration after either a single or double injury (Figure 1D and 5B). Therefore, while axons are less capable of regenerating after a double injury compared to a single injury, *tir-1* inhibits axon regeneration in multiple contexts. In response to the reviewers’ original comments, we removed the discussion of what might be responsible for the difference in regenerative ability from the previous version of the manuscript.

Genetic dependencies: TIR-1 regulates regeneration using similar mechanisms after different types of injury. We found that regeneration was increased relative to wild type in *tir-1(-)*, *nsy1(-)*, and *pmk-1(-)* animals after a single injury (Figure 5—figure supplement 1). Therefore, TIR1, NSY-1 and PMK-1 inhibit regeneration after either a single or double injury. In addition, we found that the increased regeneration phenotypes are not additive in *tir-1(-); nsy-1(-)* or *tir1(-); pmk-1(-)* double mutants compared to any of the single mutants, indicating the two MAP kinases function in the same pathway as TIR-1 to inhibit axon regeneration (Figure 5—figure supplement 1). Together with our previous data, these data reveal that TIR-1 inhibits regeneration with NSY-1 and PMK-1 in response to multiple types of injury.

Reviewer #2 (Recommendations for the authors):I commend the authors on making extensive revisions. The demonstration of a role for Tir-1 functioning through the Ask1 pathway to regulate axon regeneration is clear, although would be solidified with the genetic rescue. Unfortunately, despite the new data, the findings on Tir-1 and axon degeneration remain difficult to follow. The double cut system is of unclear physiological relevance, key experiments yield mechanistically distinct results that are not clearly addressed by the authors, and whatever Tir-1 does do for axon degeneration in *C. elegans* is quite different than its role in other characterized organisms and so the broader significance is questionable.

Thank you for your comments. We have addressed each one below.

- In Figure 2E what is the impact on the regeneration of overexpressing tir-1 in a wild-type background? Overexpressing tir-1 is only shown in a Tir-1 mutant background. However, overexpression of tir-1 may cause inhibition of regeneration even in wild type, which would then mean the data in the mutant background cannot be cleanly interpreted as a rescue.

We asked whether overexpressing *tir-1b* in wild type GABA neurons causes an additive phenotype with *tir-1(qd4)* or if it rescues the regeneration phenotype of *tir-1(qd4)* animals. We found that overexpression of *tir-1b* did not inhibit regeneration in wild type animals (Figure 2E). Therefore, we conclude that cell-autonomous expression of *tir-1b* in GABA neurons is sufficient to inhibit axon regeneration in animals that otherwise lack *tir-1*.

We have added the above data to Figure 2E.

- 2F: Is a deletion of the autoinhibitory domain of human Sarm1 being expressed? Expression of such a construct in flies or mammalian cells leads to cell death. Could this just be making the cell sick so it can't regenerate as well? What is the influence of expressing activated human Sarm on regeneration in WT?

Although ‘sick’ is difficult to define molecularly, we found that expressing a low copy array of hSarm1 lacking its autoinhibitory N-terminal, herein referred to as hSARM1_SAMTIR,_ in otherwise wild type animals resulted in wild type amounts of axon regeneration (Figure 2F). The severed proximal axon fragments remained intact, whether they did or did not regenerate. Note the wild type + *pGABA::hSARM1_SAMTIR_* strain was derived from the *tir-1(qd4)* + *pGABA::hSARM1_SAMTIR_* (Line 1) strain; therefore, they express the same extrachromosomal array. Together, our data indicate hSARM1_SAMTIR_ is sufficient to inhibit axon regeneration cell-autonomously in *C. elegans* motor neurons.

We have added the above data to Figure 2F.

- Figure 3: The key demonstration that Tir-1 promotes axon degeneration in worms is the finding that double-cut middle axon fragments degenerate slightly less in tir-1 mutant than in wild type. This key result is not rescued and should be.

Axon degeneration was restored in *tir-1(qd4)* animals that express *tir-1b* specifically in their GABA neurons. The same extrachromosomal *tir-1b* expression array has no significant effect on middle fragment degeneration in wild type animals. The ceiling effect in this assay and the use of extrachromosomal overexpression arrays prevent us from conclusively determining whether expressing multiple copies of *tir-1b* could activate more degeneration than observed in wild type animals or if a single copy of *tir-1b* in GABA neurons would restore degeneration to wild type levels. However, these data demonstrate that expressing multiple copies of *tir-1b* in GABA neurons is sufficient to restore degeneration in *tir-1(qd4)* animals.

We have added the above data to Figure 3H.

- Two recent PLos Genetics papers (Herrmann et al. and Brace et al.) demonstrated that Sarm1 Nadase activity is necessary to regulate the Ask1 MAP kinase pathway in *Drosophila*. This is in contradiction to the findings here and should be discussed. Ideally, the authors would test the requirement of the Tir-1 NADase in the classic left-right asymmetry paradigm in C elegans (known to require Ask1), but that might be beyond the scope of this paper.

As the reviewer indicates, our finding that TIR-1 regulates regeneration in coordination with NSY-1 signaling and independently of its NADase activity contrasts with recent findings that dSarm regulates neuromuscular junction formation and glial phagocytosis using both its NADase activity and interaction with Ask1 signaling (Brace et al., 2022; Herrmann et al., 2022). However, the TIR-1 mechanism does parallel the mechanism by which dSarm regulates axon transport within uninjured bystander neurons, which occurs independently of its NADase activity and with ASK1 MAP kinase signaling (Hsu et al., 2021). These data, along with the findings that dSARM regulates degeneration independently of Ask1 signaling, that TIR-1 can induce degeneration independently of its NADase activity when overexpressed and that TIR-1 can promote degeneration by antagonising DLK-1-PMK-3 MAPK, comprise only a subset of the varied interactions between dSARM/SARM1/TIR-1, their NADase activity and MAPK signaling. The mix and match nature of these interactions indicate that dSarm and TIR-1 have evolved to intricately and specifically regulate a number of diverse processes, including development, cell death, the injury response, neuronal maintenance and innate immunity using variations of common themes. We agree that determining whether TIR-1 regulates cell fate determination and its other developmental functions independently of its NADase activity is also interesting but beyond the scope of this paper.

**References**

Brace, EJ, et al. Distinct developmental and degenerative functions of SARM1 require NAD+ hydrolase activity. PLoS Genet **18**, e1010246, doi:10.1371/journal.pgen.1010246 (2022).

Ding, C, et al. Activation of the CaMKII-Sarm1-ASK1-p38 MAP kinase pathway protects against axon degeneration caused by loss of mitochondria. Elife **11**, doi:10.7554/eLife.73557 (2022).

Essuman, K, et al. The SARM1 Toll/Interleukin-1 Receptor Domain Possesses Intrinsic NAD(+) Cleavage Activity that Promotes Pathological Axonal Degeneration. Neuron **93**, 1334-1343 e1335, doi:10.1016/j.neuron.2017.02.022 (2017).

Ghosh, AS, et al. DLK induces developmental neuronal degeneration via selective regulation of proapoptotic JNK activity. J Cell Biol **194**, 751-764, doi:10.1083/jcb.201103153 (2011).

Herrmann, KA, et al. Divergent signaling requirements of dSARM in injury-induced degeneration and developmental glial phagocytosis. PLoS Genet **18**, e1010257, doi:10.1371/journal.pgen.1010257 (2022).

Horsefield, S, et al. NAD+ cleavage activity by animal and plant TIR domains in cell death pathways. Science **365**, 793-799, doi:10.1126/science.aax1911 (2019).

Hsu, JM, et al. Injury-Induced Inhibition of Bystander Neurons Requires dSarm and Signaling from Glia. Neuron **109**, 473-487 e475, doi:10.1016/j.neuron.2020.11.012 (2021).

Lezi, E, et al. An Antimicrobial Peptide and Its Neuronal Receptor Regulate Dendrite Degeneration in Aging and Infection. Neuron **97**, 125-138 e125, doi:10.1016/j.neuron.2017.12.001 (2018).

Loring, HS, et al. A phase transition enhances the catalytic activity of SARM1, an NAD(+) glycohydrolase involved in neurodegeneration. Elife **10**, doi:10.7554/eLife.66694 (2021).

Miller, BR, et al. A dual leucine kinase-dependent axon self-destruction program promotes Wallerian degeneration. Nat Neurosci **12**, 387-389, doi:10.1038/nn.2290 (2009).

Nichols, ALA, et al. The Apoptotic Engulfment Machinery Regulates Axonal Degeneration in *C. elegans* Neurons. Cell Rep **14**, 1673-1683, doi:10.1016/j.celrep.2016.01.050 (2016).

Peterson, ND, et al. Pathogen infection and cholesterol deficiency activate the *C. elegans* p38 immune pathway through a TIR-1/SARM1 phase transition. Elife **11**, doi:10.7554/eLife.74206 (2022).

Rawson, RL, et al. Axons degenerate in the absence of mitochondria in *C. elegans*. Curr Biol **24**, 760-765, doi:10.1016/j.cub.2014.02.025 (2014).

Summers, DW, et al. SARM1-specific motifs in the TIR domain enable NAD+ loss and regulate injury-induced SARM1 activation. Proc Natl Acad Sci U S A **113**, E6271-E6280, doi:10.1073/pnas.1601506113 (2016).

Summers, DW, Milbrandt, J, & DiAntonio, A. Palmitoylation enables MAPK-dependent proteostasis of axon survival factors. Proc Natl Acad Sci U S A **115**, E8746-E8754, doi:10.1073/pnas.1806933115 (2018).

Veriepe, J, Fossouo, L, & Parker, JA. Neurodegeneration in *C. elegans* models of ALS requires TIR-1/Sarm1 immune pathway activation in neurons. Nat Commun **6**, 7319, doi:10.1038/ncomms8319 (2015).

Walker, LJ, et al. MAPK signaling promotes axonal degeneration by speeding the turnover of the axonal maintenance factor NMNAT2. Elife **6**, doi:10.7554/eLife.22540 (2017).

Xiong, X, & Collins, CA. A conditioning lesion protects axons from degeneration via the Wallenda/DLK MAP kinase signaling cascade. J Neurosci **32**, 610-615, doi:10.1523/JNEUROSCI.3586-11.2012 (2012).

Yang, J, et al. Pathological axonal death through a MAPK cascade that triggers a local energy deficit. Cell **160**, 161-176, doi:10.1016/j.cell.2014.11.053 (2015).

Yanik, MF, et al. Neurosurgery: functional regeneration after laser axotomy. Nature **432**, 822, doi:10.1038/432822a (2004).